# On the Symmetries of Deep Learning Models and their Internal Representations

**Charles Godfrey**[1,*]**, Davis Brown**[1,*]**, Tegan Emerson**[1,3,4]**, Henry Kvinge**[1,2,3]
[1]Pacific Northwest National Laboratory,
[2]Department of Mathematics, University of Washington,
[3]Department of Mathematics, Colorado State University,
[4]Department of Mathematical Sciences, University of Texas, El Paso
[*]Equal contribution
`first.last@pnnl.gov`

## Abstract

Symmetry has been a fundamental tool in the exploration of a broad range of complex systems. In machine learning, symmetry has been explored in both models and data. In this paper we seek to connect the symmetries arising from the architecture of a family of models with the symmetries of that family's internal representation of data. We do this by calculating a set of fundamental symmetry groups, which we call the *intertwiner groups* of the model. Each of these arises from a particular nonlinear layer of the model and different nonlinearities result in different symmetry groups. These groups change the weights of a model in such a way that the underlying function that the model represents remains constant but the internal representations of data inside the model may change. We connect intertwiner groups to a model's internal representations of data through a range of experiments that probe similarities between hidden states across models with the same architecture. Our work suggests that the symmetries of a network are propagated into the symmetries in that network's representation of data, providing us with a better understanding of how architecture affects the learning and prediction process. Finally, we speculate that for ReLU networks, the intertwiner groups may provide a justification for the common practice of concentrating model interpretability exploration on the activation basis in hidden layers rather than arbitrary linear combinations thereof.

## 1 Introduction

Symmetry provides an important path to understanding across a range of disciplines. This principle is well-established in mathematics and physics, where it has been a fundamental tool (e.g., Noether's Theorem [Noe18]). Symmetry has also been brought to bear on deep learning problems from a number of directions. There is, for example, a rich research thread that studies symmetries in data types that can be used to inform model architectures. The most famous examples of this are standard convolutional neural networks which encode the translation invariance of many types of semantic content in natural images into a network's architecture. In this paper, we focus on connections between two other types of symmetry associated with deep learning models: the symmetries in the learnable parameters of the model and the symmetries across different models' internal representation of the same data.

The first of these directions of research starts with the observation that in modern neural networks there exist models with different weights that behave identically on all possible input. We show in section 3 that at least some of these equivalent models arise because of symmetries intrinsic to the

36th Conference on Neural Information Processing Systems (NeurIPS 2022).

nonlinearities of the network. We call these groups of symmetries, each of which is attached to a particular type of nonlinear layer $\sigma$ of dimension $n$, the *intertwiner groups* $G_{\sigma_n}$ of the model. These intertwiner groups come with a natural action on network weights for which the realization map to function space of [JGH18] is invariant (proposition 3.4). As such they provide a unifying framework in which to discuss well-known weight space symmetries such as permutation of neurons [Bre+19] and scale invariance properties of ReLU and batch-norm [IS15; NH10].

Next, we tie our intertwiner groups to the symmetries between different model's internal representations of the same data. We do this through a range of experiments that we describe below; each builds on a significant recent advance in the field.

**Neural stitching with intertwiner groups:** The work of [BNB21; Csi+21; LV15] demonstrated that one can take two trained neural networks, say A and B, with the same architecture but trained from different randomly initialized weights, and connect the early layers of network A to the later layers of network B with a simple "stitching" layer and achieve negligible loss in prediction accuracy. This was taken as evidence of the similarity of strong model's representations of data. Though the original experiments use a fully connected linear layer to stitch, we provide theoretical evidence in theorem 4.2 that much less is needed. Indeed, we show that the intertwiner group (which has far fewer parameters in general) is the minimal viable stitching layer to preserve accuracy. We conduct experiments stitching networks at ReLU activation layers with the stitching layer restricted to elements of the group $G_{\text{ReLU}}$ showing in fig. 1 that one can stitch CNNs on CIFAR-10 [Kri09] with only elements of $G_{\text{ReLU}}$ incurring less than $\approx 10\%$ accuracy penalty at most activation layers. This is surprisingly close to the losses found when one allows for a much more expressive linear layer to be used to stitch two networks together. However, we see that there remains a significant gap between the stitching accuracies obtained using $G_{\text{ReLU}}$ and fully connected linear layers; this provides independent confirmation of earlier findings that neurons of networks trained with different random seeds (i.e. with independent initializations and different random batches) are not simply permutations of each other [Li+15; Wan+18]. It is also consistent with observed phenomena such as distributed representations in hidden features [GBC16, §15.4] and perhaps also polysemantic neurons [Ola+20].

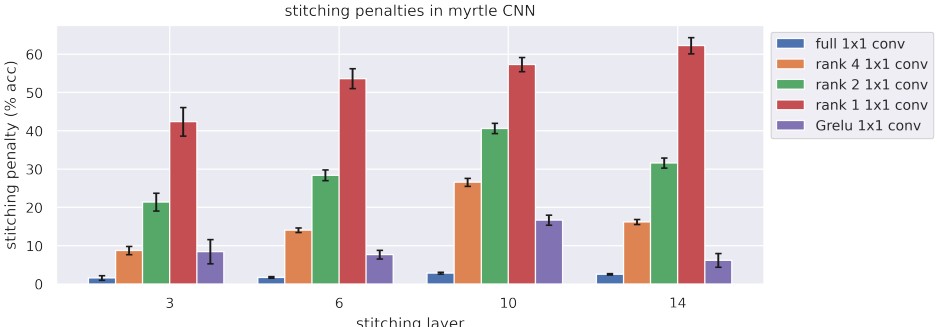

Figure 1: Full, reduced rank and $G_{\text{ReLU}}$ 1-by-1 convolution stitching penalties (4.3) for Myrtle CNNs [Pag18] on CIFAR-10. Confidence intervals were obtained by evaluating stitching penalties for 32 pairs models trained with different random seeds. The accuracy of the models was $91.3 \pm 0.2\%$.

**Representation dissimilarity measures for $G_{\text{ReLU}}$:** In section 5 we present two statistical dissimilarity measures, $G_{\text{ReLU}}$-Procrustes and $G_{\text{ReLU}}$-CKA, for ReLU-activated hidden features in different networks, say $A$ and $B$. Our measures are counterparts of orthogonal Procrustes distance (see e.g. [DDS21b]) and Centered Kernel Alignment (CKA)[1] [Kor+19] respectively, which are invariant to orthogonal transformations, and are maximized when the hidden features of networks A and B agree up to orthogonal transformations. In contrast $G_{\text{ReLU}}$-Procrustes and $G_{\text{ReLU}}$-CKA are invariant to $G_{\text{ReLU}}$ transformations, and are maximized when hidden features agree up to $G_{\text{ReLU}}$ transformations. We compare and contrast our measures with their orthogonal counterparts, as well as with stitching experiment results. Figure 2 shows a comparison of $G_{\text{ReLU}}$ and orthogonal CKA measures.

---

[1]with linear or RBF kernel.

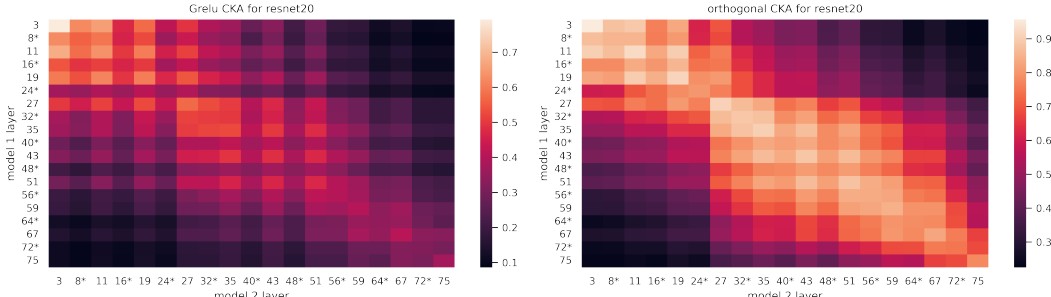

Figure 2: $G_{\mathrm{ReLU}}$-CKA and orthogonal CKA between layers of two ResNet20s trained on CIFAR-10. Results averaged over 16 pairs of models trained with different random seeds . Layers marked with '*' occur inside residual blocks (remark 3.6). For further details see section 5.

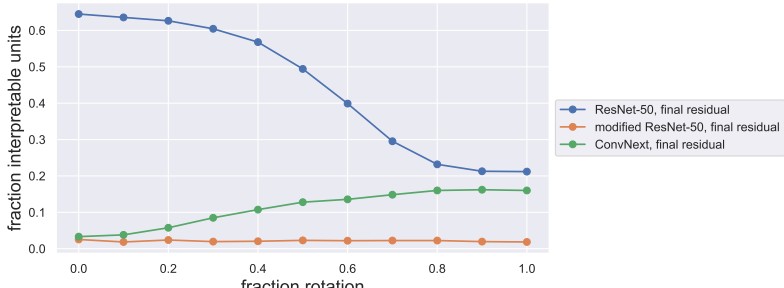

Figure 3: Fraction of network dissection interpretable units under rotations of the representation basis for a ResNet-50, as well as a modified ResNet-50 and a ConvNeXt model (both without an activation function on the residual output). Section 6 contains details and further discussion.

**Impact of activation functions on interpretability:** An intriguing finding from [Bau+17, §3.2] was that individual neurons are more interpretable than random linear combinations of neurons. Our results on intertwiner groups (theorem 3.3) predict that this is a particular feature of ReLU networks. Indeed, fig. 3 shows that in the absence of ReLU activations interpretability does not decrease when one moves from individual neurons to linear combinations of neurons — section 6 describes this experiment in further detail. This result suggests that intertwiner groups provide a theoretical justification for the explainable AI community's focus on individual neuron activations, rather than linear combinations thereof [Erh+09; Na+19; Yos+15; ZF14; Zho+15], but that this justification is only valid for layers with certain activation functions.

Taken together, our experiments provide evidence that a network's symmetries (realized through intertwiner groups) propagate down to symmetries of a model's internal representation of data. Since understanding how different models process the same data is a fundamental goal in fields such as explainable AI and the safety of deep learning systems, we hope that our results will provide an additional lens under which to examine these problems.

## 2 Related work

The research on symmetries of neural networks is extensive, hence we aim to provide a representative sample knowing it will be incomplete. [Bre+19; FB17; GBC16; Yi+19] study the effect of weight space symmetries on the loss landscape. On the other hand, [BMC15; GBC16; Kun+21; Men+19] study the effect of weight space symmetries on training dynamics, while [GBC16; RK20] show that weight space symmetries pose an obstruction to model identifiability.

Neural stitching was introduced as a means of comparing learned representations between networks in [LV15]. In [BNB21] it was shown to have intriguing connections with the "Anna Karenina " (high performance models share similar internal representations of data) and "more is better" (stitching

later layers of a weak model to early layers of a model trained with more data/parameters/epochs can improve performance) phenomena. [Csi+21] considered constrained stitching layers by restricting the rank of the stitching matrix or by introducing an $\ell_1$ sparsity penalty. Our methods are distinct in that we explicitly optimize over the intertwiner group for ReLU nonlinearities (permutations and scalings). Both [BNB21; Csi+21] compare their stitching results with statistical dissimilarity measures such as CKA. Our $G_{\text{ReLU}}$-Procrustes measure is a close relative of the permutation Procrustes distance introduced in [Wil+21], and our $G_{\text{ReLU}}$-CKA is a an instance of CKA [Kor+19] in which the kernel is taken to be $\max\{x_1 \cdot y_1, \ldots, x_d \cdot y_d\}$.

[Li+15] developed algorithms for obtaining a permutation to align neurons, and [Wan+18] introduced *neuron activation subspace matching* and used it to study similarity of hidden feature representations. [AHS22; Ent+22; Tat+20] all aligned neurons with permutations with the goal of obtaining low-loss paths between the weights of networks. The objectives for neuron alignment used in these works include maximizing correlation ([AHS22; Li+15; Tat+20]), maximizing a "match" (as defined in [Wan+18]), simulated annealing search algorithms ([Ent+22]), direct alignment of weights via a bilinear assignment problem and a "straight-through estimator" of back-propagated training loss ([AHS22]). Each of these is distinct from our method, which explicitly seeks a permutation minimizing training loss and searches for one using standard convex relaxation methods for permutation optimization.

Approaches to deep learning interpretability sometimes assume that the activation basis is special [Erh+09; Na+19; Yos+15; ZF14; Zho+15]. Studying individual neurons rather than linear combinations of neurons significantly reduces the complexity of low-level approaches to the understanding of neural networks [Elh+21]. In tension with this, many different projections of hidden layer activations appear to be semantically coherent [Sze+14]. However, [Bau+17] found evidence that the hidden feature vectors closer to the coordinate basis align more with human concepts than vectors sampled uniformly from the unit sphere.

## 3    The symmetries of nonlinearities

Let $\mathrm{Mat}_{n_1,n_0}(\mathbb{R})$ be the algebra of all $n_0 \times n_1$ real matrices and $GL_n(\mathbb{R})$ be the group of all invertible $n \times n$ matrices. Let $\sigma : \mathbb{R} \to \mathbb{R}$ be a continuous function. For any $n \in \mathbb{N}$, we can build a *nonlinearity* $\sigma_n$ from $\mathbb{R}^n$ to $\mathbb{R}^n$ by applying $\sigma$ coordinatewise, i.e., $\sigma_n(x_1, \ldots, x_n) = (\sigma(x_1), \ldots, \sigma(x_n))$. Fix some $k > 1$ and for each $1 \leq i < k$ let $\ell_i : \mathbb{R}^{n_{i-1}} \to \mathbb{R}^{n_i}$ be the composition of an affine layer and a nonlinear layer, so that $\ell_i(x) := \sigma_{n_i}(W_i x + b_i)$, and let $\ell_k(x) := W_k x + b_k$. Here $W_i \in \mathrm{Mat}_{n_i,n_{i-1}}(\mathbb{R})$ and $b_i \in \mathbb{R}^{n_i}$ are the weights and bias of layer $i$ respectively. We define $f : \mathbb{R}^{n_0} \to \mathbb{R}^{n_k}$ to be the neural network $f = \ell_k \circ \cdots \circ \ell_1$. For each $1 \leq i < k - 1$ we can then decompose $f$ as $f = f_{>i} \circ f_{\leq i}$ where

$$f_{\leq i} = \ell_i \circ \cdots \circ \ell_1 \quad \text{and} \quad f_{>i} = \ell_k \circ \cdots \circ \ell_{i+1}.$$

We define

$$W := (W_i, b_i \,|\, i = 1, \ldots, k) \quad \text{and} \quad \mathcal{W} := \prod_{i=1}^{k} (\mathrm{Mat}_{n_i,n_{i-1}}(\mathbb{R}) \times \mathbb{R}^{n_i})$$

where the former is the collection of all weights of $f$ and the latter is the space of all possible weights for a given architecture. When we want to emphasize the dependence of $f$ on weights $W$, we write $f(-, W)$ (and similarly $f_{\leq i}(-, W), f_{>i}(-, W)$).

One of the topics this work will consider is vector space bases for $f$'s hidden spaces $\mathbb{R}^{n_i}$, for $1 \leq i \leq k - 1$. We will investigate the legitimacy of analyzing features $f_{\leq i}(D)$ for dataset $D \subset \mathbb{R}^{n_0}$ with respect to the *activation basis* for $\mathbb{R}^{n_i}$ which is simply the usual coordinate basis, $e_1, \ldots, e_{n_i}$ where $e_j = [\delta_{j\ell}]_{\ell=1}^{n_i}$ is naturally parameterized by individual neuron activations. Note that $\mathbb{R}^{n_i}$ has an infinite number of other possible bases that could be chosen.

### 3.1    Intertwiner Groups

For any $0 \leq i < k$, elements of $GL_{n_i}(\mathbb{R})$ can be applied to the hidden activation space $\mathbb{R}^{n_i}$ both before and after the nonlinear layer $\sigma_{n_i}$. We define

$$G_{\sigma_{n_i}} := \{A \in GL_{n_i}(\mathbb{R}) \,|\, \text{ there exists a } B \in GL_{n_i}(\mathbb{R}) \text{ such that } \sigma_{n_i} \circ A = B \circ \sigma_{n_i}\}.$$

| Activation | $G_{\sigma_n}$ | $\phi_\sigma(A)$ |
|---|---|---|
| $\sigma(x) = x$ (identity) | $GL_n(\mathbb{R})$ | $A$ |
| $\sigma(x) = \frac{e^x}{1+e^x}$ | $\Sigma_n$ | $A$ |
| $\sigma(x) = \mathrm{ReLU}(x)$ | Matrices $PD$, where $D$ has positive entries | $A$ |
| $\sigma(x) = \mathrm{LeakyReLU}(x)$ | Same as ReLU as long as negative slope $\neq 1$ | $A$ |
| $\sigma(x) = \frac{1}{\sqrt{2\pi}}e^{-\frac{x^2}{2}}$ (RBF) | Matrices $PD$, where $D$ has entries in $\{\pm 1\}$ | $\mathrm{abs}(A)$ |
| $\sigma(x) = x^d$ (polynomial) | Matrices $PD$, where $D$ has non-zero entries | $A^{\odot d}$ |

Table 1: Explicit descriptions of $G_{\sigma_n}$ and $\phi_\sigma$ for six different activations. Here $P \in \Sigma_n$ is a permutation matrix, $D$ is a diagonal matrix, abs denotes the entrywise absolute value, and $A^{\odot d}$ denotes the entrywise $d$th power.

Informally, we can understand $G_{\sigma_{n_i}}$ to be the set of all invertible linear transformations whose action on $\mathbb{R}^{n_i}$ prior to the nonlinear layer $\sigma_{n_i}$ has an equivalent invertible transformation after $\sigma_{n_i}$. This is an instance of the common procedure of understanding a function by understanding those operators that commute with it. For any $A \in GL_{n_i}(\mathbb{R})$, we can write $\sigma(A)$ for the $n_i \times n_i$ matrix formed by applying $\sigma$ to all entries in $A$.

**Lemma 3.1.** *Suppose $\sigma(I_n)$ is invertible and for each $A \in GL_n(\mathbb{R})$ define $\phi_\sigma(A) = \sigma(A)\sigma(I_n)^{-1}$. Then $G_{\sigma_n}$ is a group, $\phi_\sigma : G_{\sigma_n} \to GL_n(\mathbb{R})$ is a homomorphism and $\sigma_n \circ A = \phi_\sigma(A) \circ \sigma_n$.*

We defer all proofs to appendix E. We include concrete examples of $\sigma$ for small $n_i$ there as well.

**Definition 3.2.** When the hypotheses of lemma 3.1 are satisfied (namely, $\sigma(I_n)$ is invertible) we call $G_{\sigma_n}$ the **intertwiner group of the activation** $\sigma_n$. We denote the image of the homomorphism $\phi_\sigma$ as $\phi_\sigma(G_{\sigma_n})$.

The intertwiner group $G_{\sigma_n}$ and $\phi_\sigma$ are concretely described for a range of activations in table 1 — the last two examples motivate the generality of definition 3.2. Note also that in both of those cases $A \mapsto \phi_\sigma(A)$ is *not* a homomorphism on all of $GL_n(\mathbb{R})$, but *is* a homomorphism when restricted to the appropriate subgroup $G_{\sigma_n}$. While a substantial part of table 1 can be found scattered in prior work, our calculations in appendix E.2 deal with the different cases of table 1 in a uniform way, by what amounts to an algorithm that compute $G_{\sigma_n}$ and $\phi_\sigma$ given in terms of elementary properties of any (reasonable) activation function $\sigma$.[2] As design of activation functions remains an active industry (for example [Elh+22]), our techniques for computing $G_{\sigma_n}$ could be useful in future studies of network symmetries.

The following theorem shows that the activation basis is intimately related to the intertwiner group of ReLU: $G_{\mathrm{ReLU}}$ admits a natural group-theoretic characterization in terms of the rays spanned by the activation basis, and dually the rays spanned by the activation basis can be recovered from $G_{\mathrm{ReLU}}$. While both its statement and proof are elementary, our interest in this theorem lies in the question of whether it could *potentially* provide theoretical justification for focusing model interpretation studies on individual activations. We investigate this question further in section 6.

**Theorem 3.3.** *The group $G_{\mathrm{ReLU}_n}$ is precisely the stabilizer of the set of rays $\{\mathbb{R}_{\geq 0}e_i \subset \mathbb{R}^n | i = 1, \ldots, n\}$. Moreover if $\mathbb{R}_{\geq 0}v_1, \ldots, \mathbb{R}_{\geq 0}v_N \subseteq \mathbb{R}^n$ is a finite set of rays stabilized by $G_{\mathrm{ReLU}}$, then for each $v_i = [v_{i1}, \ldots, v_{in}]^T$, it must be that $v_{ij} = 0$ for all but one $j \in \{1, \ldots, n\}$. Equivalently up to multiplication by a positive scalar every $v_i$ is of the form $\pm e_j$ for some $j$.*

## 3.2 Weight space symmetries

The intertwiner group is also a natural way to describe the weight space symmetries of a neural network. We denote by $\mathcal{F} \subseteq C(\mathbb{R}^{n_0}, \mathbb{R}^{n_k})$ the space of continuous functions that can be described by a network with the same architecture as $f$. As described in [JGH18] there is a **realization map** $\Phi : \mathcal{W} \to \mathcal{F}$ mapping weights $W \in \mathcal{W}$ to the associated function $f \in \mathcal{F}$. $\Phi$ arises because there are generally multiple sets of weights that yield the same function. We will show that $\Phi$ is invariant

---

[2]We defer further discussion of and references to this prior work to appendix E.2.

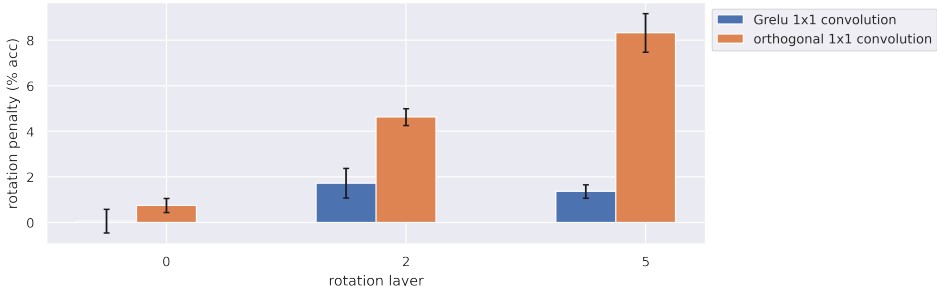

Figure 4: Rotation penalties for Myrtle CNNs on the CIFAR-10 dataset. Confidence intervals were obtained by performing 10 independent trials of the experiment with different random seeds, and baseline accuracy was $\approx 87\%$..

with respect to an action of the intertwiner groups on $\mathcal{W}$ so that intertwiner groups form a set of "built-in" weight space symmetries of $f$. This result, which encompasses phenomena including permutation symmetries of hidden neurons, is well known in many particular cases (e.g., [GBC16, §8.2.2], [Bre+19, §3], [FB17, §2], [Men+19, §3], [RK20, §3, A]). From Proposition 3.4 we can also derive corollaries regarding symmetries of the loss landscape — these are included in appendix E.6.

**Proposition 3.4.** *Suppose $A_i \in G_{\sigma_{n_i}}$ for $1 \leq i \leq k-1$, and let*

$$W' = (A_1 W_1, A_1 b_1, A_2 W_2 \phi_\sigma(A_1^{-1}), A_2 b_2, \dots, W_k \phi_\sigma(A_{k-1}^{-1}), b_k)$$

*Then, as functions, for each $m$*

$$f_{\leq m}(x, W') = \phi_\sigma(A_m) \circ f_{\leq m}(x, W) \text{ and } f_{>m}(x, W') = f_{>m}(x, W) \circ \phi_\sigma(A_m)^{-1}, \quad (3.5)$$

*In particular, $f(x, W') = f(x, W)$ for all $x \in \mathbb{R}^{n_0}$. Equivalently, we have $\Phi(W') = \Phi(W) \in \mathcal{F}$.*

*Remark* 3.6. We will show in appendix E that the statement of this theorem must be modified if the architecture of $f$ contains residual connections. By placing suitable restrictions on the matrices $A_i$[3] we can recover a form of eq. (3.5) provided $m$ occurs at the end of a residual block. However, there doesn't seem to be a way to obtain such an identity when $m$ occurs *inside* a residual block; we see empirical evidence consistent with this point in figs. 2, 6 and 9 below.

### 3.3 A "sanity test" for intertwiners

To test proposition 3.4 with a simple experiment, we begin with a Myrtle CNN [Pag18] network[4] trained for 50 epochs on the CIFAR-10 dataset, fix a *pre-activation layer* $l$, and apply a transformation $A$ to the weights $W_l$ and biases $b_l$ to obtain $AW_l$ and $Ab_l$ (we only act on channels, hence in practice this is implemented by an auxiliary 1-by-1 convolution layer). We consider 2 choices of $A$: (i) a random element of $G_{\text{ReLU}}$, where $P$ is a random permutation and the diagonal entries of $D$ are sampled from a lognormal distribution, and (ii) a random orthogonal matrix, obtained as the "$Q$" in a $QR$-decomposition of a random matrix $X$ with independent standard normal entries.

Next, we freeze layers up to and including $l$ and finetune the later layers for another 50 epochs. We refer to the difference between the validation accuracy before and after applying the transformation $A$ and finetuning as a **rotation penalty**. Based on proposition 3.4, when $A \in G_{\text{ReLU}}$ the network should be able to recover reasonable performance even with the transformed features — for example, by updating $W_{l+1}$ to $W_{l+1}\phi_\sigma(A)^{-1}$. On the other hand, with probability 1 there is no matrix $B$ such that updating $W_{l+1}$ to $W_{l+1}B$ counteracts the effect of an orthogonal rotation $A$ on all possible input. We see that this is indeed the case in fig. 4: transforming by $A \in G_{\text{ReLU}}$ rather a random orthogonal matrix results in significantly smaller rotation penalties.

## 4 Intertwining group symmetries and model stitching

In this section we provide evidence that some of the differences between distinct model's internal representations can be explained in terms of symmetries encoded by intertwiner groups. We do this

---

[3]Namely, that $A_l = A_m$ if layers $l$ and $m$ are joined by a sequence of residual connections.

[4]This is a simple 5-layer CNN, with no residual connections described further in appendix D.

using the stitching framework from [BNB21; Csi+21], and begin by reviewing the concept of network stitching.

Suppose $f, \tilde{f}$ are two networks as in section 3, with weights $W, \tilde{W}$ respectively. For any $1 \leq l \leq k-1$ we may form a **stitched network** $S(f, \tilde{f}, l, \varphi) : \mathbb{R}^{n_0} \to \mathbb{R}^{n_k}$ defined in the notation of section 3 by $S(f, \tilde{f}, l, \varphi) = \tilde{f}_{>l} \circ \varphi \circ f_{\leq l}$ – here $\varphi : \mathbb{R}^{n_l} \to \mathbb{R}^{n_l}$ is a **stitching layer**. In a typical stitching experiment one trains networks $f$ and $\tilde{f}$ from different initializations and freezes their weights, constrains $\varphi$ to some simple function class $\mathcal{S}$ (e.g., affine maps in [BNB21]), and trains $S(f, \tilde{f}, l, \varphi)$ by optimizing $\varphi$ alone. The final validation accuracy $\mathrm{Acc}\, S(f, \tilde{f}, l, \varphi)$ of $S(f, \tilde{f}, l, \varphi)$ is then considered a measure of similarity (or lack therof) of the internal representations of $f$ and $\tilde{f}$ in $\mathbb{R}^{n_l}$ — in this framework the situation

$$\mathrm{Acc}\, S(f, \tilde{f}, l, \varphi) \approx \mathrm{Acc}\, f, \mathrm{Acc}\, \tilde{f} \qquad (4.1)$$

corresponds to high similarity since the hidden representations of model $f$ and $\tilde{f}$ could be related by a transformation $\mathcal{S}$.

Recall that even though the networks $f(W)$ and $f(W')$ may be *equal as functions*, their hidden representations need not be the same (an example of this is given in appendix C). Our next result shows that in the case where $f$ and $\tilde{f}$ do only differ up to an element of $G_{\sigma_{n_l}}$, eq. (4.1) is achievable even when the stitching function class $\mathcal{S}$ is restricted down to elements of $\phi_\sigma(G_{\sigma_{n_l}})$ (see definition 3.2).

**Theorem 4.2.** *Suppose* $\tilde{W} = (A_1 W_1, A_1 b_1, A_2 W_2 \phi_\sigma(A_1^{-1}), A_2 b_2, \ldots, W_k \phi_\sigma(A_k^{-1}), b_k)$ *where* $A_i \in G_{\sigma_{n_i}}$ *for all* $i$. *Then eq.* (4.1) *is achievable with equality if the stitching function class* $\mathcal{S}$ *containing* $\varphi$ *contains* $\phi_\sigma(G_{\sigma_{n_l}})$.

Motivated by theorem 4.2, we attempt to stitch various networks at ReLU activation layers using the group $G_{\mathrm{ReLU}}$ described in Figure 1. Every matrix $A \in G_{\mathrm{ReLU}}$ can be written as $PD$, where $P$ is a permutation matrix and $D$ is diagonal with positive diagonal entries — hence optimization over $G_{\mathrm{ReLU}}$ requires optimizing over permutation matrices. We use the well-known convex relaxation of permutation matrices to doubly stochastic matrices and describe our optimization procedure in greater detail in D.2.

Figure 1 gives the difference between the average test error of Myrtle CNN networks $f$ and $\tilde{f}$ and the network $S(f, \tilde{f}, l, \varphi)$, which we call the **stitching penalty**:

$$\frac{\mathrm{Acc}(f) + \mathrm{Acc}(\tilde{f})}{2} - \mathrm{Acc}(S(f, \tilde{f}, l, \varphi)). \qquad (4.3)$$

In our experiments $S(f, \tilde{f}, l, \varphi)$ was stitched together at layer $l$ via a stitching transformation $\varphi$ that was either optimized over all affine transformations, reduced rank affine transformations as in [Csi+21] or transformations restricted to $G_{\mathrm{ReLU}}$. We consider only the ReLU activation layers, as these are the only layers where the theory of section 3 applies, and we only act on the channel tensor dimension — in practice, this is accomplished by means of 1-by-1 convolution operations. In particular, with $G_{\mathrm{ReLU}}$ we are *only permuting and scaling channels*. Lower values indicate that the stitching layer was sufficient to translate between the internal representation of $f$ at layer $l$ and the internal representation of $\tilde{f}$.

We find that when we learn a stitching layer over arbitrary affine transformations of channels, we can nearly achieve the accuracy of the original models. When we only optimize over $G_{\mathrm{ReLU}}$ there is an appreciable increase in test error difference. This is consistent with findings in [Csi+21; Li+15; Wan+18] discussed in section 2, and also consistent with observations that hidden features of neural networks exhibit distributed representations and polysemanticism [Ola+20]. Nonetheless, that $S(f, \tilde{f}, l, \varphi)$ is able get within less than $10\%$ of the accuracy of $f$ and $\tilde{f}$ in all but one layer suggests that elements of $G_{\mathrm{ReLU}}$ can account for a substantial amount of the variation in the internal representations of independently trained networks. We include the reduced rank transformations as the dimension of their parameter spaces is greater than that of $G_{\mathrm{ReLU}}$, and yet they incur significantly higher stitching penalties. If $n_l$ is the number of channels, we have $\dim G_{\mathrm{ReLU}_{n_l}} = n_l$ whereas the dimension of rank $r$ transformations is $2n_l \cdot r - r^2$ (hence greater than $\dim G_{\mathrm{ReLU}_{n_l}}$ even for

$r = 1$).[5] Finally, in the specific case of the Myrtle CNNs the stitching penalties incurred when using any layer other than 1-by-1 convolution with a rank 1 matrix all follow similar trends: they increase up to the third activation layer, then decrease at the final activation layer.

Further stitching results on the ResNet20 architecture can be found in appendix D.3, including an experiment where we modify the architecture to have LeakyReLU activation functions, vary the negative slope of the LeakyReLU, and find similar stitching penalties up to but not including a slope of 1. This result is consistent with our calculations in table 1, where we find that for any LeakyReLU negative slope $\neq 1$ the intertwiner is the same as $G_{\mathrm{ReLU}}$ (when the negative slope is 1, LeakyReLU$(x) = x$ and so the intertwiner is all of $\mathrm{GL}_n$).

## 5 Dissimilarity measures for the intertwiner group of ReLU

Stitching penalties can be viewed as task oriented measures of hidden feature dissimilarity. From a different perspective, we can consider raw statistical measures of hidden feature dissimilarity. In the design of measures of dissimilarity, a crucial choice is the group of transformations under which the dissimilarity measure is invariant. For example, Centered Kernel Alignment (CKA) [Kor+19] with the dot product kernel is invariant with respect to orthogonal transformations and isotropic scaling. We ask for a statistical dissimilarity metric $\mu$ on datasets $X, Y \in \mathbb{R}^{N \times d}$ with the properties that (0) $0 \leq \mu(X, Y) \leq 1$, (i) ($G_{\mathrm{ReLU}}$-*Invariance*) If $A, B \in G_{\mathrm{ReLU}_d}$ and $v, w \in \mathbb{R}^d$ then $\mu(XA + \mathbf{1}v^T, YB + \mathbf{1}w^T) = \mu(X, Y)$, and (ii) (*Alignment Property*) $\mu(X, Y) = 1$ if ($*$) $Y = XA + \mathbf{1}v^T$ for some $A \in G_{\mathrm{ReLU}_d}, v \in \mathbb{R}^d$. To motivate this question, we note that given such a metric $\mu$, one can detect if $X$ and $Y$ do *not* differ by an element of $G_{\mathrm{ReLU}}$ by checking if $\mu(X, Y) < 1$. Our basic tool for ensuring (i) is the next lemma.

**Lemma 5.1.** *Suppose* $\mu(XA, YB) = \mu(X, Y)$ *if* $A, B$ *are* either *positive diagonal matrices or permutation matrices. Then, (i) holds.*

In effect, this allows us to divide the columns of $X$ and $Y$ by their norms to achieve invariance to the action of positive diagonal matrices and then apply dissimilarity measures for the permutation group such as those presented in [Wil+21]. Ensuring (ii) seems to require case-by-case analysis to determine an appropriate normalization constant.

**Definition 5.2** ($G_{\mathrm{ReLU}}$-Procrustes). Let $D_X = \mathrm{diag}(|X_{[:,i]}|)$ and $D_Y = \mathrm{diag}(|Y_{[:,i]}|)$. Assuming these are invertible, let $\tilde{X} = X D_X^{-1}$ and $\tilde{Y} = Y D_Y^{-1}$. Let $\delta$ be the permutation Procrustes distance between $\tilde{X}, \tilde{Y}$, defined by $\delta := \min_{P \in \Sigma_d} |\tilde{X} - \tilde{Y}P|$ (as pointed out in [Wil+21] this can be computed via the linear sum assignment problem). Then the $G_{\mathrm{ReLU}}$-**Procrustes measure** is

$$\mu_{\text{Procrustes}}(X, Y) := 1 - \frac{\delta}{2\sqrt{d}}.$$

The factor of $2\sqrt{d}$ ensures this lies in $[0, 1]$, and equals 1 if (and only if) the condition $*$ of (ii) holds.

|  | layer 3 | layer 6 | layer 10 | layer 14 |
|---|---|---|---|---|
| $G_{\mathrm{ReLU}}$ | $0.6208 \pm 0.008$ | $0.5106 \pm 0.005$ | $0.4432 \pm 0.004$ | $0.4899 \pm 0.002$ |
| Orthogonal | $0.7724 \pm 0.028$ | $0.5743 \pm 0.040$ | $0.5087 \pm 0.016$ | $0.5825 \pm 0.019$ |

Table 2: $G_{\mathrm{ReLU}}$ and orthogonal Procrustes similarities for Myrtle CNNs trained on CIFAR-10. Confidence intervals were obtained by evaluating similarities for 32 pairs models trained with different random seeds.

We apply $G_{\mathrm{ReLU}}$-Procrustes and orthogonal Procrustes similarities to 4 different hidden representations from Myrtle CNNs in table 2 and many more layers of ResNet20s in fig. 9, all trained on CIFAR-10 [Kri09]. In keeping with the discussion of section 4, we only consider permutations or orthogonal transformations of channels (for details on how this is implemented we refer

---

[5]A valid concern is that the preceding analysis underestimates the size of $G_{\mathrm{ReLU}}$ by ignoring a large discrete factor: $G_{\mathrm{ReLU}_{n_l}}$ has $n_l!$ connected components. In appendix E.7 we carry out a comparison of the sizes of the parameter spaces of $G_{\mathrm{ReLU}_{n_l}}$ and reduced rank transformations inspired by the machinery of $\epsilon$-nets, obtaining the same conclusion that even the space of rank 1 transformations is larger than $G_{\mathrm{ReLU}}$.

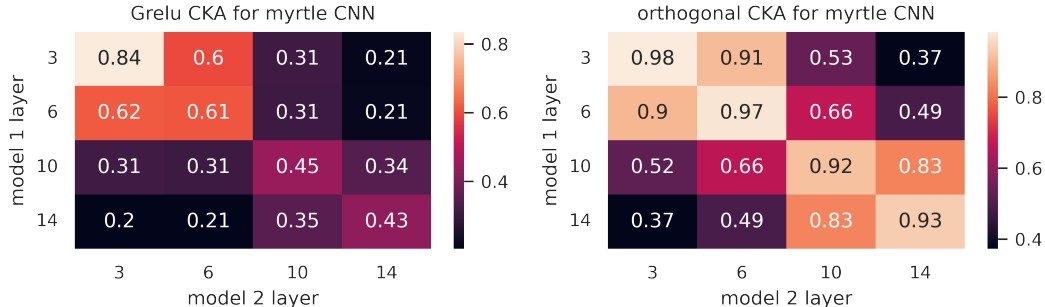

Figure 5: $G_{\text{ReLU}}$-CKA and orthogonal CKA for two Myrtle CNNs with different random seeds trained on CIFAR-10. Results averaged over 16 such pairs of models .

to appendix D.7). We see that distinct representations register less similarity in terms of $G_{\text{ReLU}}$-Procrustes than they do in terms of orthogonal Procrustes. This makes sense as similarity up to $G_{\text{ReLU}}$-transformation requires a greater degree of absolute similarity between representations than is required of similarity up to orthogonal transformation (the latter being a higher-dimensional group containing all of the permutations in $G_{\text{ReLU}}$). Otherwise patterns in $G_{\text{ReLU}}$-Procrustes similaritiy largely follow those of orthogonal Procrustes, with similarity between representations decreasing as one progresses through the network, only to increase again in the last layer. This correlates with the stitching penalties of fig. 1, which increase with depth only to decrease in the last layer.

**Definition 5.3** ($G_{\text{ReLU}}$-CKA). Assume that $X$ and $Y$ are data matrices that have been centered by subtracting means of rows: $X \leftarrow X - \frac{1}{d}\mathbf{1}\mathbf{1}^T X$ and $Y \leftarrow Y - \frac{1}{d}\mathbf{1}\mathbf{1}^T Y$. Let $\tilde{X} = XD_X^{-1}$ and $\tilde{Y} = YD_Y^{-1}$. Let $\tilde{x}_1, \ldots, \tilde{x}_N$ be the rows of $\tilde{X}$, and similarly for $\tilde{Y}$. Form the matrices $K, L \in \mathbb{R}_{\geq 0}^{N \times N}$ defined by $K_{ij} = \max(\tilde{x}_i \odot \tilde{x}_j)$ and $L_{ij} = \max(\tilde{y}_i \odot \tilde{y}_j)$ where $\odot$ is the Hadamard product. Then the $G_{\text{ReLU}}$-CKA for $X$ and $Y$ is defined as:

$$\mu_{\text{CKA}}(X, Y) := \frac{\text{HSIC}_1(K, L)}{\sqrt{\text{HSIC}_1(K, K)}\sqrt{\text{HSIC}_1(L, L)}}. \tag{5.4}$$

where $\text{HSIC}_1$ is the unbiased form of the Hilbert-Schmidt independence criterion of [NRK21, eq. 3].

Symmetry of the $\max$ function ensures (i), the Cauchy-Schwarz inequality ensures $\mu_{\text{CKA}}(X, Y) \in [0, 1]$, and we claim that $\mu_{\text{CKA}}(X, Y) = 1$ if the condition $*$ of (ii) is met. We do not claim 'if and only if', however we point out the following in lemma E.44: if $A$ is a matrix such that $\max(Ax_1 \odot Ax_2) = \max(x_1 \odot x_2)$ for all $x_1, x_2 \in \mathbb{R}^d$, then $A$ is of the form $PD$ where $P$ is a permutation matrix and $D$ is diagonal with diagonal entries in $\{\pm 1\}$. In fact, $\mu_{\text{CKA}}$ is simply an instance of CKA for a the "max kernel."

**Lemma 5.5.** *The function* $\kappa : \mathbb{R}^d \times \mathbb{R}^d \to \mathbb{R}$ *defined by* $k(x, y) = \max(x \odot y)$ *is a positive semi-definite kernel.*

As with CKA [Kor+19], this metric makes sense even if $X, Y$ are datasets in $\mathbb{R}^d, \mathbb{R}^{d'}$ respectively with $d \neq d'$. Results for a pair of Myrtle CNNs trained on CIFAR-10 with different random seeds, as well as standard orthogonal CKA for comparison, are shown in fig. 5. Analogous results for ResNet20s are shown in fig. 2. We find that $G_{\text{ReLU}}$-CKA respects basic trends found in their orthogonal counterparts: model layers at the same depth are more similar, early layers are highly similar, and the metric surfaces the block structure of the ResNet in fig. 2 (layers inside residual blocks are less similar than those at residual connections). One notable difference for $G_{\text{ReLU}}$-CKA in figs. 2 and 5 is that the similarity difference between early and later layers in the orthogonal CKA (discussed for ResNets in [Rag+21]) shown in (b) is less pronounced in (a), and in fact later layers are found to be less similar between runs. We found similar results for stitching in figs. 1 and 6.

## 6 Interpretability of the coordinate basis

In this section we explore the confluence of model interpretability and intertwiner symmetries using *network dissection* from [Bau+17]. Network dissection measures alignment between the individual

neurons of a hidden layer and single, pre-defined concepts (see appendix F.1 for the methodology). We adapt an experiment from [Bau+17] to compare the axis-aligned interpretability of hidden activation layers with and without an activation function. Bau et al. compares the interpretability of individual neurons, measured via network dissection, with that of random orthogonal rotations of neurons. We likewise rotate the hidden layer representations and then measure their interpretability. Using the methodology from [Dia05], we define a random orthogonal transform $Q$ drawn uniformly from $SO(n)$ by using Gram-Schmidt to orthonormalize the normally-distributed $QR = A \in \mathbb{R}^{n^2}$. Like in [Bau+17], we also consider smaller rotations $Q^\alpha \in SO(n)$ where $0 \leq \alpha \leq 1$, where $\alpha$ is chosen to form a minimal geodesic rotating from $I$ to $Q$. [Bau+17] found that the number of interpretable units decreased away from the activation basis as $\alpha$ increased for layer5 of an AlexNet.

We compare three models trained on ImageNet: a ResNet-50, a modified ResNet-50 where we remove the ReLU on the residual outputs (training details in appendix F.3), and a ConvNeXt [Liu+22] analog of the ResNet-50, which also does not have an activation function before the final residual output. We give results in fig. 3, and provide sample unit detection outputs and full concept labels for the figures in appendix F.2. As was shown in [Bau+17], interpretability decreases as we rotate away from the axis for the normal ResNet-50 in appendix F.3. On the other hand, with no activation function, neuron interpretability does not drop with rotation for the modified ResNet-50 and the ConvNeXt. We note that the models without residual activation functions also have far fewer concept covering units for a given basis. Interestingly, while the number of interpretable units remains constant for the residual output of the modified ResNet-50, for the ConvNeXt model it actually *increases*. We find similar results, where the number of interpretable units increase with rotation, for the convolutional layer inside the residual block for the modified ResNet-50 in fig. 19.

## 7 Limitations

Our theoretical analysis in section 3 does not account for standard regularization techniques that are known to have symmetry-breaking effects (for example weight decay reduces scaling symmetry). More generally, we do not account for any implicit regularization of our training algorithms. As illustrated in figs. 1 and 6, stitching with intertwiner groups appears to have significantly more architecture-dependent behaviour than stitching with arbitrary affine transformations (however, since different architectures have different symmetries this is to be expected). Our empirical tests of the dissimilarity measures in section 5 are limited to what [Kor+19] terms "sanity tests"; in particular we did not perform the specificity, sensitivity and quality tests of [DDS21a].

## 8 Conclusion

In this paper we describe groups of symmetries that arise from the nonlinear layers of a neural network, calculate these symmetry groups for a number of different types of nonlinearities, and explore their fundamental properties and connection to weight space symmetries. Next, we provide evidence that these symmetries induce symmetries in a network's internal representation of the data that it processes, showing that previous work on the internal representations of neural networks can be naturally adapted to incorporate awareness of the intertwiner groups that we identify. Finally, in the special case where the network in question has ReLU nonlinearities, we find experimental evidence that intertwiner groups justify the special place of the activation basis within interpretable AI research.

## 9 Acknowledgements

This research was supported by the Mathematics for Artificial Reasoning in Science (MARS) initiative at Pacific Northwest National Laboratory. It was conducted under the Laboratory Directed Research and Development (LDRD) Program at at Pacific Northwest National Laboratory (PNNL), a multiprogram National Laboratory operated by Battelle Memorial Institute for the U.S. Department of Energy under Contract DE-AC05-76RL01830.

The authors would also like to thank Nikhil Vyas for useful discussions related to this work.

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
