# Supplementary Material for:
# On the Symmetries of Deep Learning Models and their Internal Representations

**Charles Godfrey**[1,*], **Davis Brown**[1,*], **Tegan Emerson**[1,3,4], **Henry Kvinge**[1,2,3]
[1]Pacific Northwest National Laboratory,
[2]Department of Mathematics, University of Washington,
[3]Department of Mathematics, Colorado State University,
[4]Department of Mathematical Sciences, University of Texas, El Paso
[*]Equal contribution
first.last@pnnl.gov

## A   Societal Impact

Though deep learning models are in the process of being deployed for safety critical applications, we still have very little understanding of the structure and evolution of their internal representations. In this paper we discuss one aspect of these representations. We hope that by better illuminating the inner workings of these networks, we will be a small part of the larger effort to make deep learning more understandable, reliable, and fair.

## B   Code availability

Our code can be found at https://github.com/pnnl/modelsym.

## C   Examples

We first give an example of two networks with distinct weights which are functionally equivalent. Let $f$ be a 2 layer network with $\mathrm{ReLU}$ activations and weight matrices

$$W_1 = \begin{bmatrix} 1 & 0 \\ 0 & 2 \end{bmatrix} \quad \text{and} \quad W_2 = \begin{bmatrix} 3 & 0 \\ 0 & 1 \end{bmatrix}$$

(and biases = 0). Let $\tilde{f}$ be a network with the same architecture, but with weights

$$W_1 = \begin{bmatrix} 0 & 2 \\ 1 & 0 \end{bmatrix} \quad \text{and} \quad W_2 = \begin{bmatrix} 0 & 3 \\ 1 & 0 \end{bmatrix}.$$

Then one can verify that $\tilde{f}(x) = f(x)$ for all $x \in \mathbb{R}$, but that the weights of $f$ and $\tilde{f}$ differ.

We also work through a small example of $\phi_{\sigma_n}$ where $n = 2$. Assume that $\sigma$ is the ReLU nonlinearity. Then,

$$A = \begin{bmatrix} 0 & 1 \\ 2 & 0 \end{bmatrix}$$

belongs to $G_{\sigma_2}$, and we can compute directly that

$$\mathrm{ReLU} \circ \begin{bmatrix} 0 & 1 \\ 2 & 0 \end{bmatrix} \begin{bmatrix} x_1 \\ x_2 \end{bmatrix} = \begin{bmatrix} \mathrm{ReLU}(x_2) \\ \mathrm{ReLU}(2x_1) \end{bmatrix} = \begin{bmatrix} \mathrm{ReLU}(x_2) \\ 2\,\mathrm{ReLU}(x_1) \end{bmatrix},$$

36th Conference on Neural Information Processing Systems (NeurIPS 2022).

where in the last equality we used the fact that $\mathrm{ReLU}(ax) = a\,\mathrm{ReLU}(x)$ when $a$ is positive. On the other hand,

$$\begin{bmatrix} 0 & 1 \\ 2 & 0 \end{bmatrix} \circ \mathrm{ReLU}(\begin{bmatrix} x_1 \\ x_2 \end{bmatrix}) = \begin{bmatrix} 0 & 1 \\ 2 & 0 \end{bmatrix} \begin{bmatrix} \mathrm{ReLU}(x_1) \\ \mathrm{ReLU}(x_2) \end{bmatrix} = \begin{bmatrix} \mathrm{ReLU}(x_2) \\ 2\,\mathrm{ReLU}(x_1) \end{bmatrix}.$$

## D   Experimental Details

In this section we provide additional experimental results, as well as implementation details for the purposes of reproducibility. All experiments were run on Nvidia GPUs using PyTorch [Pas+19].

### D.1   Sampling pairs of models trained with different random seeds

We began by training 100 models with different random seeds (i.e. with independent initializations and different random batches) for each of the following architectures:

(i) Myrtle CNN: a simple 5-layer feed-forward CNN with batch normalization.[6]

(ii) ResNet20: a ResNet tailored to the CIFAR-10 dataset (numbers of channels are $16, 32, 64$ respectively in the 3 residual blocks).

(iii) ResNet18: an ImageNet-style ResNet adapted to the input size of CIFAR-10 — much wider than the above (numbers of channels are $64, 128, 256$ respectively in the 3 residual blocks).

More detailed architecture schematics are included in figs. 26a, 27a and 28a.

All models were trained for 50 epochs using the Adam optimizer with PyTorch's default settings. We use a batch size of 32, initial learning rate $0.001$ and 4 evenly spaced learning rate drops with factor $0.5$. We augment data with translations of up to 2 pixels (padded as necessary with the mean RGB value for CIFAR-10) and left-right flips, and we save the weights with best validation accuracy. In the rotation penalties experiment of fig. 4 the fine-tuning stage uses the same hyperparameters as the initial training phase (though of course only a subset of parameters recieve gradient updaates during fine-tuning). Training this many CIFAR-10 models on a reasonable budget of time and computing resources was greatly aided by the excellent FFCV library [Lec+22].

In the later stitching and dissimilarity measure experiments, we sample pairs of models from these "zoos" uniformly with replacement (but of course making sure that the two models in the pair are distinct). Thus the cost of training hundreds of models is amortized across many runs of stitching and dissimilarity measurement; this can be also viewed as bootstrap estimation of our experimental quantities of interest using empirical samples from certain distributions of CIFAR-10 models.

### D.2   Stitching Experiments

For stitching layers, we train for 20 epochs with batch size 32 and learning rate $0.001$ (with no drops), however we use vanilla SGD with no momentum (we found the approximate second-order and/or momentum aspects of Adam interacted in complicated ways with the PGD algorithm described in appendix D.2.1 below, even after following some helpful advice from the Internet[7]). Augmentation is described in the previous paragraph.

We parameterize reduced rank 1-by-1 convolutions as a composition of 2 1-by-1 convolutions, with `in_channels, out_channels = in_channels, rank` and `rank, in_channels` respectively. In contrast to [BNB21] we omit both batch norm and bias from stitching layers (to stick closely to the statement of theorem 4.2).

### D.2.1   Approximate Optimization over Permutation Matrices

By far the most complicated stitching layer is the one using $G_{\mathrm{ReLU}}$, which we describe here. Recall that $G_{\mathrm{ReLU}}$ is equal to the $n \times n$ matrices of the form $PD$, where $P \in \Sigma_n$ is a permutation matrix

---

[6]With the exception of the rotation penalties experiment in fig. 4, where we omitted batch normalization to adhere closely to the theoretical framework of section 3

[7]https://datascience.stackexchange.com/questions/31709/adam-optimizer-for-projected-gradient-descent

and $D$ is a diagonal matrix with positive entries We parameterize $D$ simply as $D = \text{diag}(\lambda_i)$ where $\lambda_1, \ldots, \lambda_{n_l} \in \mathbb{R}_{\geq 0}$ — we preserve non-negativity during training by a projected gradient descent step $D \leftarrow \text{ReLU}(D)$. During stitching layer training, we parameterize $P$ as a doubly stochastic matrix, that is, an element of the Birkhoff polytope

$$\mathcal{B} = \{A = (a_{ij}) \in \text{Mat}_{n_l, n_l}(\mathbb{R}) \, | \, a_{ij} \geq 0 \text{ for all } i, j, \mathbf{1}^T A = \mathbf{1}^T \text{ and } A\mathbf{1} = \mathbf{1}\}$$

— after each gradient descent step we project $P$ back onto $\mathcal{B}$ by the operation $P \leftarrow \text{ReLU}(P)$ followed by $P \leftarrow \text{sink}(P)$, where "sink" denotes Sinkhorn iterations. These consist of $T$ iterations of

$$A \leftarrow A \, \text{diag}(\mathbf{1}^T A)^{-1} \text{ followed by } A \leftarrow \text{diag}(A\mathbf{1})^{-1} A$$

(it is a theorem of Sinkhorn that this sequence converges to a doubly stochastic matrix of the form $DAE$ with $D, E$ positive diagonal matrices [Sin64]). We use $T = 16$ in all experiments (this choice drew on the work of [Men+18]). In addition, we add a regularization term $-\alpha|P|_2$ to the stitching objective, where $\alpha > 0$ is a hyperparameter (the motivation here is that permutation matrices are precisely the elements of $\mathcal{B}$ with maximal $\ell_2$-norm). Unless stated otherwise in our experiments $\alpha = 0.1$. We did experiment with choosing $\alpha$ by cross validation and found the particular choice of $\alpha$ was not crucial; see appendix D.5 for further details.

At evaluation time, we threshold $P$ to an actual permutation matrix via the Hungarian algorithm (specifically its implementation in scipy.optimize.linear_sum_assignment [Vir+20]). This amounts to

$$P_{\text{eval}} = \arg \max_{Q \in \Sigma_{n_l}} \text{tr}(P_{\text{train}} Q^T)$$

As stated above, we train for 20 epochs with batch size 32 and learning rate 0.001 (with no drops), using SGD with momentum 0.9. However, we allow the permutation factor to get a "head start" by keeping $D$ fixed at the identity $I$ for the first 10 epochs. This is probably not essential, as shown in appendix D.5.

Finally, before evaluating the stitched model on the CIFAR-10 validation set, we perform a no-gradient epoch *on the training data* with stitching layer $P_{\text{eval}}$. This is critical as it allows the batch normalization running means and variances in later layers to adapt to the thresholded permutation matrix $P_{\text{eval}}$; observe that if we omitted this step, during evaluation the "batch normalization layers" would not even be performing batch normalization per se, since their running statistics would be computed from features produced by a layer $P_{\text{train}}$ no longer in use.

As an aside, we also experimented with the differeniable relaxation of permutation matrices SoftSort [PE20]. Our final results were comparable, however this method took far longer ($> 10\times$) to optimize than the Birkhoff polytope method. It is perhaps of interest that we used SoftSort on permutations far larger than those of [PE20] (e.g., the 512 channels of late layers of our Myrtle CNN). The next section (appendix D.2.2) contains some of our technical findings.

We wish to acknowledge a couple articles, [Fog+13] and [LW14], that provided us with useful backround on optimization over doubly stochastic matrices.

### D.2.2  Stitching with SoftSort

We parameterized $D$ simply as $D = \text{diag}(e^{\lambda_i})$ where $\lambda_1, \ldots, \lambda_{n_l} \in \mathbb{R}$. During stitching layer training, we parameterized $P$ using SoftSort [PE20], a continuous relaxation of permutation matrices given by the formula

$$P = \text{SoftSort}(s, \tau) := \text{softmax}\left(-\frac{1}{\tau}(\text{sort}(s)\mathbf{1}^T - \mathbf{1}s^T)\right), \text{ where } s \in \mathbb{R}^{n_l},$$

$\text{sort}(s)$ denotes $s$ sorted in descending order, and $\text{softmax}$ is applied over rows. The parameter $\tau > 0$ controls $\text{softmax}$ temperature, and we were only able to obtain reasonable results when tuning it according to $\tau \approx 1/n_l$. At validation time, we threshold $P$ to an actual permutation matrix by applying $\arg \max$ over rows as in [PE20].

### D.3  Stitching and $G_{\text{ReLU}}$-dissimilarity measures for ResNets

Here we include further results for ResNet20 and ResNet18 architectures. Figure 6 and fig. 7 include results for full 1-by-1 convolution, reduced randk 1-by-1 convolution and $G_{\text{ReLU}}$ 1-by-1 convolutions

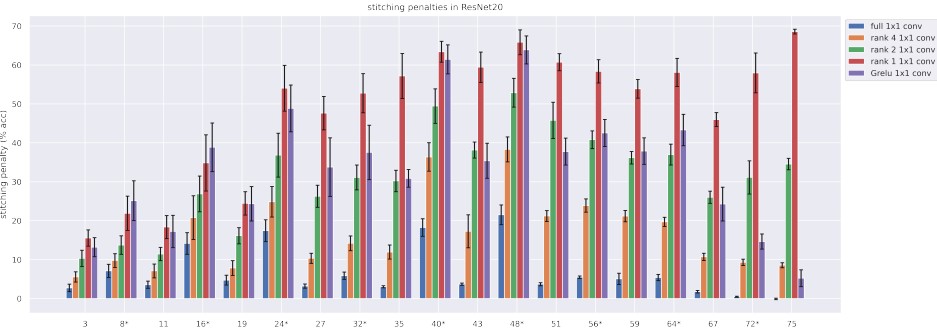

Figure 6: Full/reduced rank and $G_{\mathrm{ReLU}}$ 1-by-1 convolution stitching penalties (4.3) for ResNet20s on CIFAR-10. Confidence intervals were obtained by evaluating stitching penalties for 16 pairs of models trained with different random seeds. Accuracy of the models was $89.9 \pm 0.2$ %. Layers marked with '*' occur inside residual blocks (remark 3.6).

stitching in the ResNet20 and ResNet18 architectures respectively. Note that in general, layers inside residual blocks incur higher penalties, consisent with remark 3.6. This holds even in the full 1-by-1 convolution case, a finding that to the best of our knowledge is new.

In the case of ResNet20 we also observe that the relative ranking of the different stitching constraints tends to change inside of residual blocks: whereas $G_{\mathrm{ReLU}}$ stitching consistently outperforms rank 1 (and sometimes rank 2) stitching outside residual blocks, it consistently underperforms all strategies inside residual blocks. Lastly, we remark that the ResNet20 is significantly narrower than the Myrtle CNN (channels are 16, 32, 64 vs. 64, 128, 256, see figs. 27a and 28a), and hence the low-rank transformations account for a larger *proportion* of the available total rank (for example, in early layers of the ResNet20 rank 4 is $0.25 \cdot \mathrm{fullrank}$ whereas in the early layers of the Myrtle CNN rank 4 is $0.0625 \cdot \mathrm{fullrank}$). Heuristically, in the narrower network low-rank transformations may suffice to align for a larger fraction of the principal components of hidden features.

We also observe generally lower stitching penalties in the ResNet18 with the exception of the penultimate inside-a-residual-block layer — we do not have a satisfactory explanation for random chance performance at that layer. We also remark that while the penalties in fig. 6 are significantly higher than those in fig. 1, especially in later layers, we also saw significant dissimilarity in fig. 2 (a), especially in later layers.

We also modify the ResNet20 to use the LeakyReLU activation function and train models with different negative slopes $s$. The accuracy for two models trained with different random seeds at different LeakyReLU is given in table 3. We perform $G_{\mathrm{ReLU}}$ stitching in fig. 8. Note that for a negative slope $s = 1$, the activation function is the identity. We find the results difficult to interpret due to the significant decrease in CIFAR-10 accuracy for larger $s$. With this being said, unlike for $s << 1$, we note that the stitching penalties for $s = 1$ (and to a lesser extent, $s = 0.9$) are mostly constant throughout the layers of the network. This is most prominent for the final two ResNet20 layers (72 and 75), where the stitching penalty for models with small LeakyReLU slopes is the lowest.

Table 3: ResNet20 with LeakyReLU CIFAR-10 accuracy

| | LeakyReLU SLOPE | | | | | | |
|---|---|---|---|---|---|---|---|
| | 1e−4 | 1e−3 | 1e−2 | 0.1 | 0.5 | 0.9 | 1.0 |
| % acc. | $89.3 \pm 0.2$ | $89.4 \pm 0.2$ | $89.2 \pm 0.2$ | $89.4 \pm 0.1$ | $86.6 \pm 0.1$ | $73.0 \pm 0.2$ | $41.8 \pm 0.1$ |

Figure 9 contains $G_{\mathrm{ReLU}}$ and orthogonal Procrustes dissimilarities for the ResNet20. The 2 measures seem qualitatively quite similar in this case. For the most part the same applies to the ResNet18 in fig. 10, with the exception of layer 70 (penultimate inside-a-residual-block layer), where we see high $G_{\mathrm{ReLU}}$ *similarity*, in conflict with both fig. 7 and fig. 11 below.

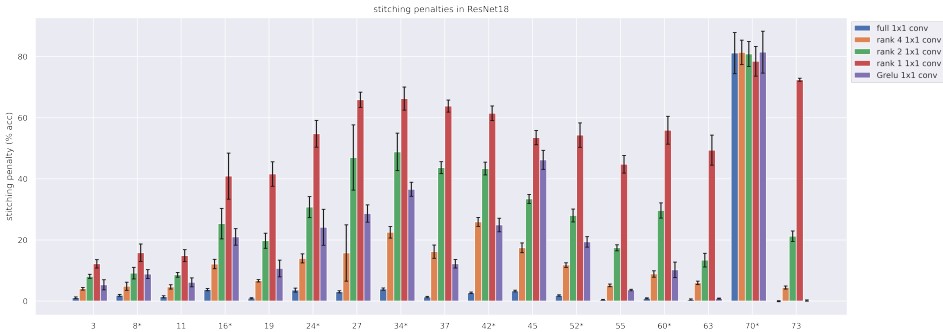

Figure 7: Full/reduced rank and $G_{\mathrm{ReLU}}$ 1-by-1 convolution stitching penalties (4.3) for ResNet18s on CIFAR-10. Confidence intervals were obtained by evaluating stitching penalties for 16 pairs of models trained with different random seeds. Accuracy of the models was $92.9 \pm 0.2$ %. Layers marked with '*' occur inside residual blocks (remark 3.6).

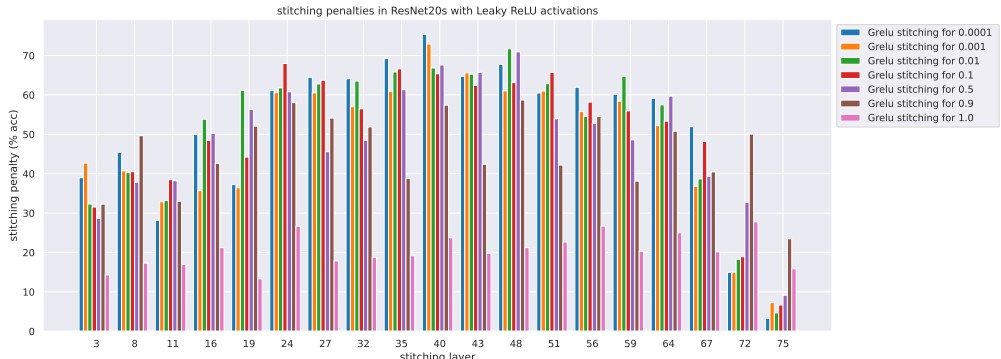

Figure 8: Stitching penalties (4.3) for ResNet20s trained with different random seeds on CIFAR-10, where respective ResNet20 models are trained with LeakyReLU activation functions with different slopes. Accuracy of the models with different LeakyReLU slopes is given in table 3.

We include $G_{\mathrm{ReLU}}$ and orthogonal CKA dissimilarities for the wider ResNet18 in fig. 11. For the most part the qualitative remarks on fig. 2 apply here as well — note also the extreme dissimilarity in layer 70 (in both $G_{\mathrm{ReLU}}$ and orthogonal cases) consistent with fig. 7.

### D.4    Stitching for a Vision Transformer

Here we include an additional stitching experiments with vision transformers from [Has+21] trained on CIFAR-10. Figure 12 include results for linear stitching and $G_{\mathrm{ReLU}}$ stitching after each transformer encoder layer. The large stitching penalties for $G_{\mathrm{ReLU}}$ are expected due to the lack of activation functions after the linear (feedforward) layers for each encoder layer.

We train 10 Compact Convolutional Transformers with sinusoidal positional encodings and six transformer blocks. The average model accuracy was $98\%$ using the distributed training-from-scratch recipe from [Has+21], which includes $6\mathrm{e}{-}2$ weight decay, augmentations (namely mixup [Zha+18] and CutMix [Yun+19]), label smoothing, and AdamW with a learning rate of $55\mathrm{e}{-}5$ with cosine scheduling.

### D.5    Choosing the negative-$\ell_2$ regularization multiplier $\alpha$ with cross validation

Here we briefly describe an experiment in which the multiplier $\alpha$ of appendix D.2.1 is chosen by cross validation. Most of the details are as in appendix D.2. However, we create a random 80-20 split of the CIFAR10 training set into a smaller training and cross-validation set. We then learn $G_{\mathrm{ReLU}}$ stitching layers for each $\alpha \in \{10^k \,|\, k = -3, -2, \ldots, 1\}$, as in appendix D.2.1, with the exception that we only optimize over our training split for 5 epochs and do not give the permutations a head start. Then,

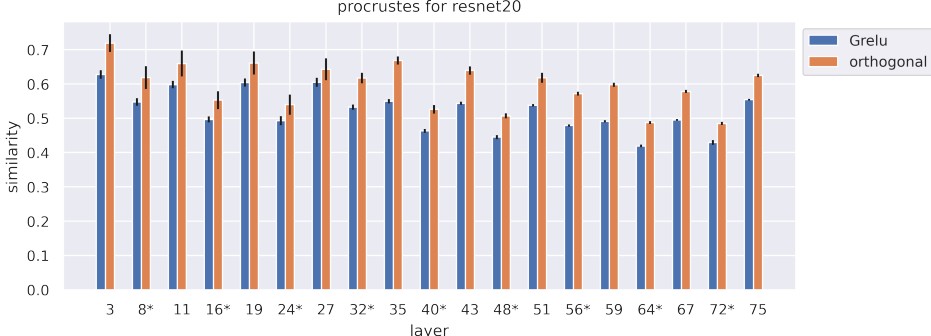

Figure 9: $G_{\text{ReLU}}$ and orthogonal Procrustes dissimilarities for two ResNet20s trained on CIFAR-10 with different random seeds. Layers marked with '*' occur inside residual blocks (remark 3.6). Confidence intervals were obtained by evaluating similarities for 32 pairs of models trained with different random seeds.

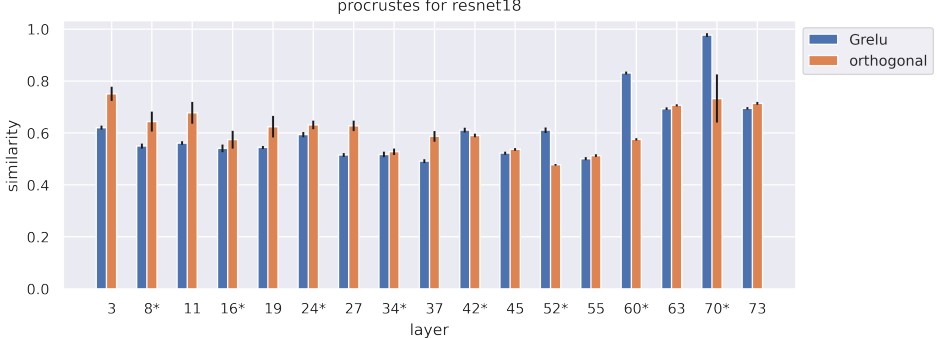

Figure 10: $G_{\text{ReLU}}$ and orthogonal Procrustes dissimilarities for two ResNet18s trained on CIFAR-10 with different random seeds. Layers marked with '*' occur inside residual blocks (remark 3.6). Confidence intervals were obtained by evaluating similarities for 32 pairs of models trained with different random seeds.

the $\alpha$ corresponding to highest accuracy on our cross validation set is selected, the corresponding model weights are loaded and we report accuracy on the regular CIFAR10 validation set. In fig. 13 we obtain very similar results to those in fig. 1. Perhaps more interestingly, in fig. 14 we see that there is substantial variance in the $\alpha$ selected by cross-validation, at all layers of our Myrtle CNN network — for reference, $\alpha = 0.1$ is used in the rest of this paper. This suggests that the particular choice of $\alpha$ is not essential to our method. Results for ResNet architectures are qualitatively similar and omitted for brevity.

### D.6 Stitching with $\ell_1$-regularized (a.k.a. LASSO) fully-connected layers

In this section we present results of a small experiment stitching with full 1-by-1 convolutional layers with $\ell_1$ penalty $\lambda|W|_1$, where $|W|_1 = \sum_{ij}|W_{ij}|$, as in [Csi+21]. We vary $\lambda \in \{0.001, 0.01, 0.1\}$ and also tried $\lambda = 1$ but found the stitching optimization to be unstable due the magnitude of the $\ell_1$ penalty (possible this could have been counteracted by decreasing the learning rate). We also record the *sparsity* of the stitching weights — if $n_l$ is the relevant channel dimension, and hence also the number of rows/columns in the square stitching matrix $W$, we measure this as

$$\frac{|\{(i,j) \in \{1,\ldots,n_l\}^2 \,|\, |W_{ij}| \leq \tau\}|}{n_l^2} \tag{D.1}$$

where $\tau$ is a threshold, in our experiments chosen to be $0.001$. Note that the sparsity of a $G_{\text{ReLU}}$ is equal to $\frac{n_l^2 - n_l}{n_l^2} = 1 - \frac{1}{n_l}$. Figure 15 illustrates the results of these experiments, and seems to show

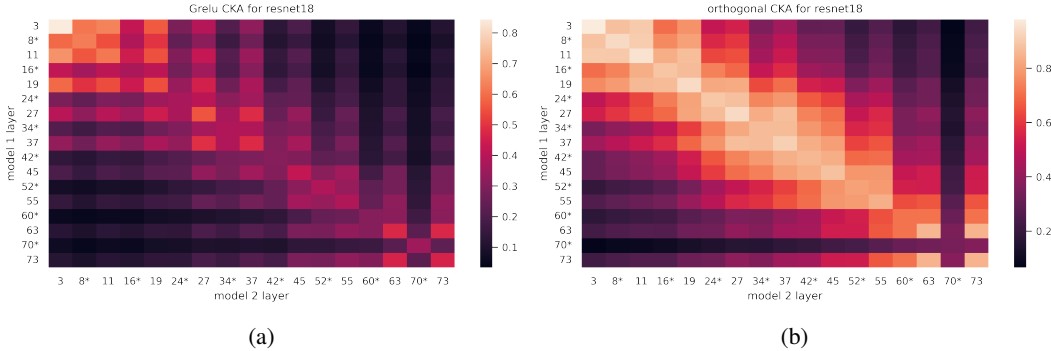

(a)                                                            (b)

Figure 11: $G_{\mathrm{ReLU}}$-CKA and orthogonal CKA for two ResNet18s trained on CIFAR-10 with different random seeds. Layers marked with '*' occur inside residual blocks (remark 3.6). Results averaged over 16 such pairs of models.

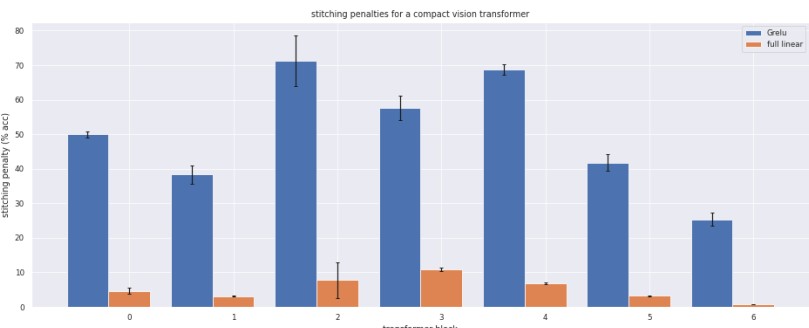

Figure 12: Linear and $G_{\mathrm{ReLU}}$ stitching penalties (4.3) for 5 pairs of vision transformers [Has+21] trained on CIFAR-10 with different random seeds. Stitching was performed after every transformer block, and notably these blocks do not end in activation functions.

that $G_{\mathrm{ReLU}}$ layers achieve low stitching penalties for their sparsity levels. Also note that in the final layer the scatter points corresponding to $G_{\mathrm{ReLU}}$ and $\lambda = 0.01$ nearly overlap.

### D.7  Implementing dissimilarity measures

As mentioned in section 5, we aim to capture invariants to permuting and scaling channels, but not spatial coordinates. This requires some care; practically speaking it means we cannot simply flatten feature vectors.

In all cases we compute our measures over the entire CIFAR-10 validation set. In particular, we do not require batched computations as in [NRK21].

#### D.7.1  Procrustes

As in [DDS21b; Wil+21]

$$\min_{P \in \Sigma_d} |\tilde{X} - \tilde{Y}P| = \sqrt{\min_{P \in \Sigma_d} |\tilde{X} - \tilde{Y}P|^2}$$

so it suffices to consider minimizing the Frobenius norm-squared, and expanding as

$$|\tilde{X} - \tilde{Y}P|^2 = |\tilde{X}|^2 + |\tilde{Y}|^2 - 2\operatorname{tr}(\tilde{X}^T\tilde{Y}P)$$

we see that this is equivalent to *maximizing* $\operatorname{tr}(\tilde{X}^T\tilde{Y}P)$. In our case $X, Y$ have shape $(N, C, H, W)$ where $N$ is the size of the entire CIFAR-10 validation set and $C, H, W$ are the channels, height, and

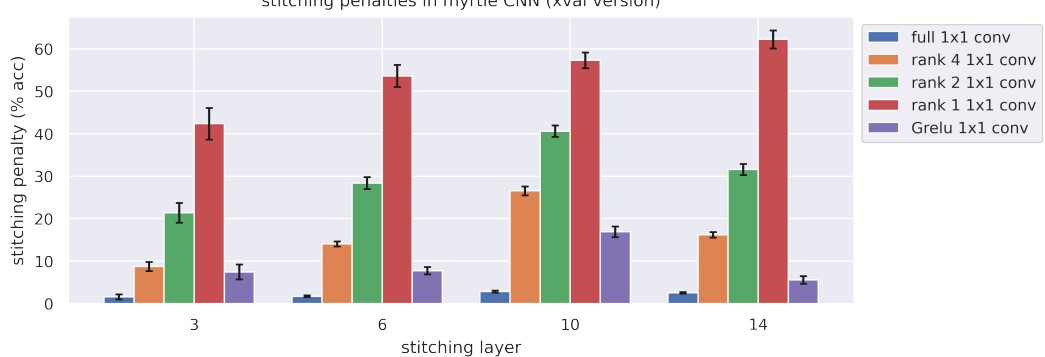

Figure 13: Full, reduced rank, and $G_{\mathrm{ReLU}}$ 1-by-1 convolution stitching penalties (4.3) for Myrtle CNNs [Pag18] on CIFAR-10, in which $\alpha$ is chosen by cross-validation. Confidence intervals were obtained by evaluating stitching penalties for 32 pairs of models trained with different random seeds. The accuracy of the models was $91.3 \pm 0.2$ %.

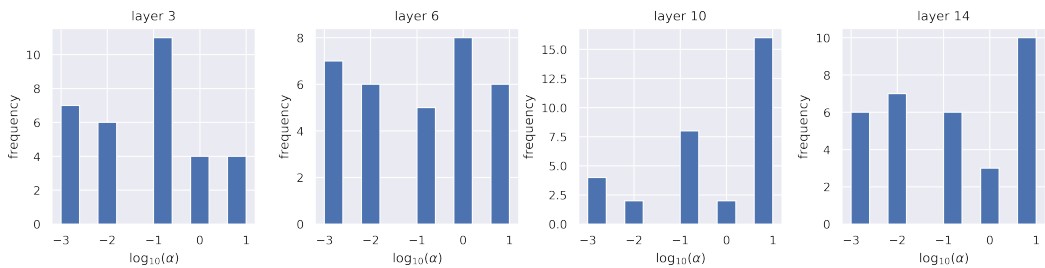

Figure 14: The histograms of $\alpha$ selected by cross validation in the experiment of fig. 13

width at the given hidden layer respectively. We want $P$ to be a $C \times C$ permutation matrix. Hence for $\tilde{X}^T \tilde{Y}$ we compute the tensor dot product

$$(\tilde{X}^T \tilde{Y})_{c,c'} = \sum_{n,h,w} \tilde{X}_{n,c,h,w} \tilde{Y}_{n,c',h,w} \tag{D.2}$$

The same method is used for orthogonal Procrustes, where instead of scipy.optimize.linear_sum_assignment we use the nuclear norm of eq. (D.2) as in [DDS21a].

### D.7.2 CKA

In this case for a set of hidden features $X$ of shape $(N, C, H, W)$ as above, we first subtract the mean over all but the channel dimension:

$$X_{n,c,h,w} \leftarrow X_{n,c,h,w} - \frac{1}{NHW} \sum_{n',h',w'} X_{n',c,h',w'}$$

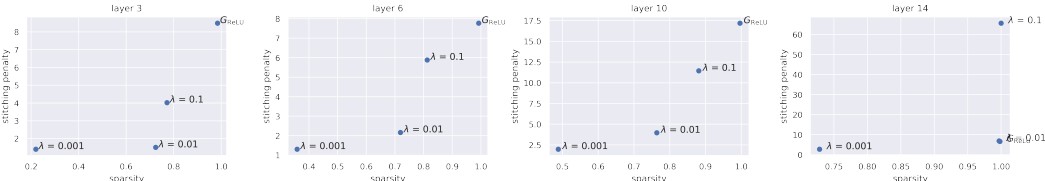

Figure 15: $\ell_1$-regularized stitching penalties versus sparsity for Myrtle CNNs, with $G_{\mathrm{ReLU}}$ stitching penalties included for comparison. Penalties and sparsities are averaged over evaluations on 32 pairs models trained with different random seeds.

and divide by the norms over all but the channel dimension:[8]

$$X_{n,c,h,w} \leftarrow \frac{X_{n,c,h,w}}{\sqrt{\sum_{n',h',w'} X_{n',c,h',w'}^2}}.$$

Next, we compute a tensor dot product of $X$ with itself *over spatial dimensions*, to obtain the shape $(N, N, C)$ tensor

$$J_{m,n,c} := \sum_{h,w} X_{n,c,h,w} X_{n,c,h,w}$$

and finally we apply $\max$ over the channel dimension to get

$$K_{m,n} = \max_c J_{m,n,c}.$$

*Remark* D.3. It could be interesting to refrain from applying a dot product over spatial dimensions, and thus measure not only similarity between hidden features of different images, but similarity between hidden features of different images *at certain locations*. However, the memory requirements would have been far beyond our computational limits.

## D.8 Dissimilarity measures for network with constant channel width

A notable feature of our plots in figs. 2, 5 and 11 is that the $G_{\text{ReLU}}$-CKA exhibits a much more significant decay with network depth than its orthogonal counterpart. From a skeptical perspective, we thought this could have something to do with dimensionality. All the networks we looked at up to this point had the feature that their channel dimension grows exponentially with depth (as seen in the last 3 figures of the appendix). When we compute the kernels $\max(\tilde{x}\_i \odot \tilde{x}\_j)$, we encounter maxima of larger and larger sets of random variables as the channel dimension increases. *If* the products inside these maxima were independent normal random variables (we are not claiming this is a reasonable heuristic), we'd expect the max to grow like $\Phi^{-1}(1 - \frac{1}{n_l})$ where $n_l$ is the channel dimension. It seemed possible that something along these lines could cause $G_{\text{ReLU}}$-CKA to drift as depth (in our experiments correlated with channel dimension) increases. Note that the dot product kernel seems comparatively immune, since (with the same heuristics of normal distribution) the expected value of $\langle \tilde{x}, \tilde{y} \rangle$ is 0 regardless of dimension.

Motivated by this train of thought, we evaluated all 4 dissimilarity measures of section 5 on a variant of our Myrtle CNN with *constant channel dimension*. The architecture of this network is identical to the one shown in fig. 26a with the exception that all channel dimensions are 512. In table 4 and fig. 16 we see that these constant width CNNs exhibit qualitatively very similar dissimilarity measures as their non-constant width counterparts. This suggests that the $G_{\text{ReLU}}$-CKA decay with network depth is *not* an artifact of increasing channel dimension.

We speculate that it's possible that the decay of $G_{\text{ReLU}}$-CKA is due to something like the *superposition hypothesis* for hidden layer features [Elh+22; Ola+20]. Roughly, in overcomplete cases where the model can use more features than basis directions in a hidden layer, it may be encoding $m$ nearly orthogonal features across $n < m$ basis directions. If this encoding is not consistent across random seeds, we expect $G_{\text{ReLU}}$-CKA to be smaller. Finally, polysemanticism may increase with depth. In a simple thought experiment, if each basis direction in layer $l$ has $a$ features encoded, layer $l + 1$ will have $2a$ features per direction if it each neuron in $l + 1$ simply sums over two neurons in $l$. Again assuming the combinations of features occuring in this polysemanticism vary accross random seeds, we would expect $G_{\text{ReLU}}$-CKA to be smaller. Simply put, superposition and polysemanticism would seem to preclude alignment of the hidden features of different networks with permutations and scaling alone.

---

[8]In retrospect, it would arguably make more sense to use standard deviation rather than $\ell_2$ norm; however, for us the choice is irrelevant in the end since the 2 choices differ by a factor of $\sqrt{NHW}$ which gets cancelled in eq. (5.4).

|              | layer 3           | layer 6           | layer 10          | layer 14          |
|--------------|-------------------|-------------------|-------------------|-------------------|
| $G_{\text{ReLU}}$  | $0.8176 \pm 0.007$ | $0.7602 \pm 0.005$ | $0.5691 \pm 0.005$ | $0.4971 \pm 0.003$ |
| orthogonal   | $0.8460 \pm 0.008$ | $0.6735 \pm 0.005$ | $0.5409 \pm 0.003$ | $0.6050 \pm 0.002$ |

Table 4: $G_{\text{ReLU}}$ and orthogonal Procrustes similarities for *constant channel width* Myrtle CNNs trained on CIFAR-10. Confidence intervals were obtained by evaluating similarities for 4 pairs of models trained with different random seeds.

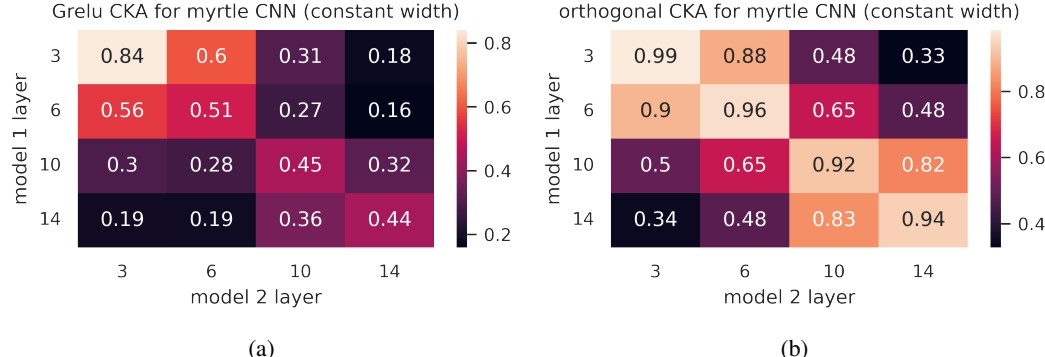

(a)                                        (b)

Figure 16: $G_{\text{ReLU}}$-CKA and orthogonal CKA for two *constant channel width* Myrtle CNNs trained on CIFAR-10 with different random seeds. Results averaged over 4 such pairs of models .

## E   Proofs

### E.1   A proof of lemma 3.1, plus some abstractions thereof

*Proof of lemma 3.1.* Since by definition $G_{\sigma_n} \subseteq GL_n(\mathbb{R})$, to prove $G_{\sigma_n}$ is a subgroup it suffices to show that if $A_1, A_2 \in G_{\sigma_n}$ then $A_1 A_2^{-1} \in G_{\sigma_n}$. By hypotheses, there are matrices $B_1, B_2 \in GL_n(\mathbb{R})$ so that

$$\sigma_n \circ A_1 = B_1 \circ \sigma_n \tag{E.1}$$
$$\text{and} \quad \sigma_n \circ A_2 = B_2 \circ \sigma_n. \tag{E.2}$$

Applying $A_2^{-1}$ on the right hand side of eq. (E.1) gives

$$\sigma_n \circ (A_1 A_2^{-1}) = B_1 \circ \sigma_n \circ (A_2^{-1}). \tag{E.3}$$

On the other hand, applying $A_2^{-1}$ on the right hand side of eq. (E.2) gives $\sigma_n = B_2 \circ \sigma_n \circ (A_2^{-1})$ and hence

$$\sigma_n \circ (A_2^{-1}) = B_2^{-1} \circ \sigma_n. \tag{E.4}$$

Combining eqs. (E.3) and (E.4) we obtain

$$\sigma_n \circ (A_1 A_2^{-1}) = B_1 B_2^{-1} \circ \sigma_n \tag{E.5}$$

and hence $G_{\sigma_n}$ is a subgroup. Next, we solve

$$\sigma_n \circ A = B \circ \sigma_n$$

for $B$ in terms of $A$ by evaluating both sides at $e_1, \ldots, e_n \in \mathbb{R}^n$ (standard basis vectors). Letting $A[:, j]$ denote the $j$-th column of $A$ we obtain

$$\sigma_n(A[:, j]) = B\sigma_n(e_j), \text{ for } j = 1, \ldots, n$$

and stacking these columns to obtain the full $n \times n$ matrix yields

$$\sigma(A) = B\sigma(I)$$

where $I \in GL_n(\mathbb{R})$ is the identity matrix and $\sigma(A)$ denotes $\sigma$ applied to the coordinates of $A$ (similarly for $\sigma(I)$). As $\sigma(I)$ is invertible by hypotheses, this implies $B = \sigma(A)\sigma(I)^{-1} =: \phi_\sigma(A)$, and finally substituting $B_i = \phi_\sigma(A_i)$ for $i = 1, 2$ in eq. (E.5) shows that

$$\sigma_n \circ (A_1 A_2^{-1}) = \phi_\sigma(A_1)\phi_\sigma(A_2)^{-1} \circ \sigma_n$$

while at the same time

$$\sigma_n \circ (A_1 A_2^{-1}) = \phi_\sigma(A_1 A_2^{-1}) \circ \sigma_n$$

so that $\phi_\sigma(A_1 A_2^{-1}) \circ \sigma_n = \phi_\sigma(A_1)\phi_\sigma(A_2)^{-1} \circ \sigma_n$. Using the invertibility of $\sigma(I)$ one more time, we conclude

$$\phi_\sigma(A_1 A_2^{-1}) = \phi_\sigma(A_1)\phi_\sigma(A_2)^{-1},$$

which implies $\phi_\sigma$ is a homomorphism. □

*Remark* E.6 (for the mathematically inclined reader). Here is a more abstract definition of $G_{\sigma_n}$ that makes lemma 3.1 appear more natural: let $X$ be a topological space with a continuous (left) action of a topological group $G$. There is a natural (right) action of $G$ on $C(X, \mathbb{R})$ by precomposition $((f, g) \mapsto f \circ g)$. For any subspace $V \subseteq C(X, \mathbb{R})$ define

$$G_V := \{g \in G \,|\, V \cdot g \subseteq V\}$$

(that is, the elements of $G$ stabilize $V$ *as a subspace*, but not necessarily pointwise — one can show this is always a subgroup of $G$). Then, for every such subspace $V$, the group $G_V$ acts linearly on $V$, and if we have a basis $f_1, \ldots, f_n \in V$, we can obtain a matrix representation of $G$ in $GL_n(\mathbb{R})$. To obtain the special case in lemma 3.1, we take $X = \mathbb{R}^n$, $G = GL_n(\mathbb{R})$ with the usual action, and $V$ to be the subspace spanned by the functions $f_i(x_1, \ldots, x_n) = \sigma(x_i)$. The condition that $\sigma(I)$ is invertible is equivalent to the condition that the column space of the matrix $(f_i(e_j))$ is $n$-dimensional, which in turn implies $V$ is $n$-dimensional.

We end this section with a lemma that allows for easy verification that $\sigma(I)$ is invertible. We used this on all the activation functions considered in table 1.

**Lemma E.7.** *Let $\sigma : \mathbb{R} \to \mathbb{R}$ be any function and let $I \in GL_n(\mathbb{R})$ be the identity matrix. Then $\sigma(I)$ is invertible provided*

$$\sigma(1) \neq \sigma(0) \text{ and } \sigma(1) \neq -(n-1)\sigma(0). \tag{E.8}$$

*Proof.* Let $N = \mathbf{1}\mathbf{1}^T - I$. Then

$$\sigma(I) = \sigma(1)I + \sigma(0)N.$$

Note that the eigenvalues of $\mathbf{1}\mathbf{1}^T$ are $n$ (corresponding to eigenvector $\mathbf{1}$) and 0's (corresponding to the orthogonal complement of $\mathbf{1}$). For any linear operator $A \in \text{Mat}_n(\mathbb{R})$ with eigenvector/eigenvalue pair $(v, \lambda)$, $v$ is easily seen to be an eigenvector of $A - I$ with eigenvalue $\lambda - 1$. Hence the eigenvalues of $N$ are $n - 1$ and $-1$, and it follows that the eigenvalues of $\sigma(I)$ are

$$\sigma(1) + \sigma(0)(n-1), \sigma(1) - \sigma(0), \ldots, \sigma(1) - \sigma(0)$$

which are all non-zero if and only if eq. (E.8) holds. □

*Remark* E.9. In particular eq. (E.8) holds when $\sigma(1) = 1, \sigma(0) = 0$ (which holds for example when $\sigma(x) = \text{ReLU}(x)$ or $\sigma(x) = x^d$). In this situation, $\sigma(I) = I$ and $\phi_\sigma(A) = \sigma(A)$ (coordinatewise application of $\sigma$). For example, if $\sigma \geq 0$ is non-negative, then $\phi_\sigma(A) = \sigma(A)$ has non-negative entries.

## E.2 Calculating intertwiner groups (for table 1)

We begin with two lemmas: the first puts a "lower bound" on $G_{\sigma_n}$ and the second is a "differential form" of the definition of the intertwiner group from section 3. Together, these two results effectively allow us to reduce calculation of intertwiner groups to the $n = 1$ case.

**Lemma E.10** (cf. [GBC16, §8.2.2], [Bre+19, §3]). *$G_{\sigma_n}$ always contains the permutation matrices $\Sigma_n$, and $\phi_\sigma$ restricts to the identity on $\Sigma_n$.*

*Proof.* If $A \in \Sigma_n$ is a permutation matrix, so that $Ae_i = e_{\pi(i)}$ where $\pi$ is a permutation of $\{1, \ldots, n\}$ then for any $x \in \mathbb{R}^n$ we observe

$$A\sigma(x) = A\left(\sum_i \sigma(x_i)e_i\right) = \sum_i \sigma(x_i)Ae_i = \sum_i \sigma(x_i)e_{\pi(i)}$$

which is exactly $\sigma$ applied coordinatewise to $\sum_i x_i e_{\pi(i)} = Ax$. □

**Corollary E.11.** *If $A \in GL_n(\mathbb{R}), P \in \Sigma_n$, and $AP \in G_{\sigma_n}$ or $PA \in G_{\sigma_n}$, then $A \in G_{\sigma_n}$.*

*Proof.* If $B = AP \in G_{\sigma_n}$, then $A = BP^{-1}$, where $B \in G_{\sigma_n}$ by hypothesis and $P \in G_{\sigma_n}$ by lemma E.10. The result follows as $G_{\sigma_n}$ is a group (lemma 3.1) and hence closed under multiplication. The other case is similar. $\square$

**Lemma E.12.** *Suppose $A, B \in GL_n(\mathbb{R})$ and $\sigma_n \circ A = B \circ \sigma_n$. Suppose $x = (x_1, \ldots, x_n)^T \in \mathbb{R}^n$ and assume $\sigma$ is differentiable at $x_1, \ldots, x_n$ as well as*

$$(Ax)_i = \sum_j a_{ij} x_j, \quad for\ i = 1, \ldots, n.$$

*Then,*

$$\mathrm{diag}(\sigma'((Ax)_i)|i = 1, \ldots, n)A = B\,\mathrm{diag}(\sigma'(x_i)|i = 1, \ldots, n)$$

*(here $\mathrm{diag} : \mathbb{R}^n \to \mathrm{Mat}_{n \times n}(\mathbb{R})$ takes a vector to a diagonal matrix). Explicitly, for each $i, j \in \{1, \ldots, n\}$*

$$\sigma'(\sum_k a_{ik} x_k) a_{ij} = b_{ij} \sigma'(x_j). \tag{E.13}$$

*Proof.* By the chain rule [Rud76, Thm. 9.15], and since the differential of a matrix is itself,

$$d\sigma_n|_{Ax} A = B d\sigma_n|_x.$$

Finally, by the definition of $\sigma_n$

$$\frac{\partial(\sigma_n(x))_i}{\partial x_j} = \frac{\partial \sigma(x_i)}{\partial x_j} = \begin{cases} \sigma'(x_j) & \text{if } i = j \\ 0 & \text{otherwise.} \end{cases}$$

$\square$

**Theorem E.14.** *Suppose $\sigma$ is non-constant, non-linear, and differentiable on a dense open set with finite complement.[9] Then,*

*(i) Every $A \in G_{\sigma_n}$ is of the form $PD$, where $P \in \Sigma_n$ and $D$ is diagonal.*

*(ii) For a diagonal $D = \mathrm{diag}(\lambda_1, \ldots, \lambda_n) \in G_{\sigma_n}$, we have $\lambda_i \in G_{\sigma_1}$ for $i = 1, \ldots, n$ and*

$$\phi_\sigma(\mathrm{diag}(\lambda_1, \ldots, \lambda_n)) = \mathrm{diag}(\phi_\sigma(\lambda_1), \ldots, \phi_\sigma(\lambda_1))$$

*where we make a slight abuse of notation: on the right hand side $\phi_\sigma$ is the homomorphism $G_{\sigma_1} \to GL_1(\mathbb{R})$.*

*In particular, $\phi_\sigma$ is determined by lemma E.10 and its behavior for $n = 1$.*

*Proof.* For any $A \in G_{\sigma_n}$ we observe that the differentiability hypotheses of lemma E.12 holds for $x \in U$ where $U$ is a dense open set with measure-0 complement. Indeed, if $t_1, \ldots, t_M \in \mathbb{R}$ are the points where $\sigma$ fails to be differentiable, we can take $U$ to be the complement of the hyperplane arangement given by

$$(\bigcup_{ij} \{x \in \mathbb{R}^n \mid x_i = t_j\}) \cup (\bigcup_{ij} \{x \in \mathbb{R}^n \mid (Ax)_i \in = t_j\}) \subseteq \mathbb{R}^n$$

Fix a row $i$ — the matrix $A$ is invertible by hypotheses, and so there must be some $j$ such that $a_{ij} \neq 0$ (otherwise the $i$-th row of $A$ is 0). For any $x \in U$, we have by lemma E.12

$$\sigma'(\sum_k a_{ik} x_k) a_{ij} = b_{ij} \sigma'(x_j) \tag{E.15}$$

and we claim that this cannot hold unless $a_{ik} = 0$ for $k \neq j$. First, there is a $(x_1, \ldots, x_n) \in U$ such that $\sigma'(x_j) \neq 0$ (otherwise $\sigma$ would be constant). Next, fixing $x_j$ at a value with $\sigma'(x_j) \neq 0$ and rearranging eq. (E.15) we have

$$\frac{\sigma'(a_{ij} x_j + \sum_{k \neq j} a_{ik} x_k)}{\sigma'(x_j)} a_{ij} = b_{ij} = \text{ constant.} \tag{E.16}$$

---

[9]The differentiability assumption is probably not necessary, however it holds in all of the examples we consider and allows us to safely use lemma E.12.

By hypothesis $\sigma$ is non-linear and so $\sigma'$ is non-constant — hence if there were some $a_{ik} \neq 0$ for $k \neq j$, the left hand side of eq. (E.16) would be non-constant.

We have shown each row of $A$ has at most one non-0 entry $a_{ij}$ and that $a_{ij} \neq 0$. For $A$ to be invertible, it must be that these non-0 entries land in distinct *columns*. This is exactly the form described in item (i).

Next, we note that for any $ij$ (without assuming $a_{ij} \neq 0$) eqs. (E.15) and (E.16) tell us

$$a_{ij} = 0 \implies b_{ij} = 0,$$

and hence if $D = \mathrm{diag}(\lambda_1, \ldots, \lambda_n) \in G_{\sigma_n}$ and $\sigma_n \circ D = E \circ \sigma_n$ (i.e. $E = \phi_\sigma(D)$), it must be that $E = \mathrm{diag}(\mu_1, \ldots, \mu_n)$ for some $\mu_1, \ldots, \mu_n \in \mathbb{R}$. Now the equation $\sigma_n \circ D = E \circ \sigma_n$ is equivalent to

$$\sigma(\lambda_i x_i) = \beta_i \sigma(x_i) \text{ for } i = 1, \ldots, n$$

which in turn is equivalent to $\lambda_i \in G_{\sigma_1}$ and $\beta_i = \phi_\sigma(\lambda_i)$ for $i = 1, \ldots, n$, proving item (ii). $\square$

In light of theorem E.14, to fill in the table of table 1 it will suffice to deal with the $n = 1$ cases, which we do below.

*Calculation of $G_{\mathrm{ReLU}}$.* We remark that this is just the "positive homogeneous" property of ReLU, which is quite well known (cf. [GBC16, §8.2.2], [FB17, §2], [Kun+21, §3], [Men+19, §3], [RK20, §3, A], [Yi+19, §2-3]). Using remark E.9 if $a \in G_{\sigma_1}$

$$\max\{0, ax\} = \max\{0, a\} \max\{0, x\}.$$

If $a < 0$ then setting $x = -1$ results in $a = 0$, a contradiction. So $a > 0$ and $\max\{0, ax\} = a \max\{0, x\}$, showing $\phi_\sigma(a) = a$. $\square$

*Modifications for* LeakyReLU. By definition, for $0 < s \ll 1$.

$$\mathrm{LeakyReLU}(x, a) := \begin{cases} sx & \text{for } x < 0 \\ x & \text{for } x \geq 0 \end{cases}$$

which we may simplify to $\mathrm{LeakyReLU}(x) = sx + (1 - s)\mathrm{ReLU}(x)$. Suppose now that

$$\mathrm{LeakyReLU}(ax) = b\,\mathrm{LeakyReLU}(x), \text{ or using our simplification}$$
$$sax + (1 - s)\mathrm{ReLU}(ax) = b(sx + (1 - s)\mathrm{ReLU}(x)). \tag{E.17}$$

If $a < 0$, we may choose $x = -1$ to obtain

$$-a = -sa - (1 - s)a = -bs \text{ and } x = 1 \text{ to obtain}$$

$$sa = b$$

showing that $a = as^2$, which is impossible when $0 < s \ll 1$. So it must be $a > 0$, and then evaluating eq. (E.17) at $x = 1$ gives $a = b$. $\square$

*The sigmoid case: $\sigma(x) = 1/(1 + e^x)$.* We will leverage of a useful fact about the sigmoid function:

$\sigma'(x)$ is a smooth probability distribution function on $\mathbb{R}$, with $\sigma'(x) > 0$ for all $x \in \mathbb{R}$. (*)

If $\sigma(ax) = b\sigma(x)$, differentiating with respect to $x$ gives

$$\sigma'(ax)a = b\sigma'(x). \tag{E.18}$$

Using eq. (*) and the fact that to probability distribution functions are proportional if and only if they are equal, we get $\sigma'(ax) = \sigma'(x)$. Then integrating from $-\infty$ to $x$ tells us $\sigma(ax)/a = \sigma(x)$; setting $x = 0$ we see $\frac{1}{2a} = \frac{1}{2}$, hence $a = 1$.

To show $\phi_\sigma = \mathrm{id}$, backtracking to eq. (E.18) and setting $x = 0$ shows $b = a$. $\square$

*The Gaussian RBF case: $\sigma(x) = \frac{1}{\sqrt{2\pi}} e^{-\frac{x^2}{2}}$.* We make use of several properties of this $\sigma(x)$:

(i) For any $a > 0$ the function $\sigma(ax)$ is a probability distribution function with mean 0 and variance $\frac{1}{a^2}$, and with $\sigma(ax) > 0$ for all $x \in \mathbb{R}$ and

(ii) $\sigma$ is an even function ($\sigma(-x) = \sigma(x)$).

Now if $\sigma(ax) = b\sigma(x)$, then the pdfs $\sigma(ax)$ and $\sigma(x)$ are proportional hence equal by item (i). Therefore they have the same means and variances — since these are $0, \frac{1}{a^2}$ and $0, 1$ respectively we conclude $a = \pm 1$.

Finally, we explain why $\phi_\sigma(A) := \text{abs}(A)$ (entrywise absolute value). Differentiating with respect to $x$ gives

$$\sigma'(ax)a = b\sigma'(x). \tag{E.19}$$

This implies $b = 1$ when $a = 1$. On the other hand differentiating item (ii) tells us $-\sigma'(-x) = \sigma'(x)$, so when $a = -1$

$$b\sigma'(x) = -\sigma'(-x) = \sigma'(x)$$

and hence $b = 1$. $\qquad\square$

*The polynomial case:* $\sigma(x) = x^d$. We remark that the description given in table 1 is implicit in [KTB19]. By theorem E.14 we only need to describe $\phi_\sigma : G_{\sigma_1} \to GL_1(\mathbb{R})$; for any $a \neq 0$

$$(ax)^d = a^d x^d$$

and this shows $G_{\sigma_1} = \mathbb{R} \setminus \{0\}$ and $\phi_\sigma(a) = a^d$. $\qquad\square$

### E.2.1 Gaussian error linear units (GeLUs)

Introduced and first studied in [HG16], these are defined as $\text{GeLU}(x) = x\Phi(x)$ where $\Phi$ is the standard normal cummulative distribution function:

$$\Phi(x) = \int_{-\infty}^{x} \frac{e^{-\frac{t^2}{2}}}{\sqrt{2\pi}} \, dt.$$

By inspecting plots in fig. 17a, we see that GeLU and ReLU are globally quite similar (they converge as $|x| \to \infty$) but that they differ when $x$ within a few standard normal standard deviations of 0. One can show that $G_{\text{GeLU}_n} = \Sigma_n$: indeed, by theorem E.14 it suffices to show that the only $\lambda \in \mathbb{R} \setminus \{0\}$ such that $\text{GeLU}(\lambda x) = \phi(\lambda)\,\text{GeLU}(x)$ for all $x$ (where $\phi$ is some non-zero function of $\lambda$) is $\lambda = 1$. Expanding, we see that

$$\lambda x \Phi(\lambda x) = \phi(\lambda)x\Phi(x), \tag{E.20}$$

and rearranging this becomes

$$\frac{\Phi(\lambda x)}{\Phi(x)} = \frac{\phi(\lambda)}{\lambda} =: c, \tag{E.21}$$

that is, the right hand side is constant as a function of $x$. Then $\Phi(\lambda x) = c\Phi(x)$, and since $\Phi$ is positive it must be $c$ is too. Moreover it must be $\lambda > 0$, as otherwise $\Phi(\lambda x)$ is monotonically decreasing while $\Phi(x)$ is increasing. Finally, letting $x \to \infty$ we see that $c = 1$, and from there we conclude $\lambda = 1$ by an argument similar to the use of item (i) in the Gaussian RBF case.

Despite the above calculation, it seems natural to ask how far $G_{\text{GeLU}}$ is from $G_{\text{ReLU}}$, in other words how badly GeLU fails to be positive homogeneous. One measure of this is obtained by letting $X \sim \mathcal{N}(0, 1)$ be a standard normal variable and computing the root-mean-square error

$$\xi(\lambda) := \sqrt{E[|\text{GeLU}(\lambda X) - \lambda \text{GeLU}(X)|^2]} \tag{E.22}$$

as a function of $\lambda > 0$, where the expectation is over $X$. Here our choice of a standard normal $X$ is motivated by the same reasoning as discissed in [HG16], namely that activation inputs are roughly standard normal, especially in the presence of batch normalization. Evaluating eq. (E.22) doesn't seem particularly tractible analytically, but it does simplify to

$$\xi(\lambda) = \lambda\sqrt{E[|x(\Phi(\lambda x) - \Phi(x))|^2]}. \tag{E.23}$$

In fig. 17b we estimate $\xi(\lambda)$ by sampling $X$ and replacing the expectation with the corresponding average. Evidently, as $\lambda \to \infty$ the function $\xi(\lambda)$ becomes linear: since $\Phi(\lambda x) \to \mathbf{1}_{x \geq 0}$ as $\lambda \to \infty$ (here $\mathbf{1}_{x \geq 0}$ is the indicator of $x > 0$, also known as the Heaviside or unit-step function), the asymptotic slope is $\sqrt{E[|x(\mathbf{1}_{x \geq 0} - \Phi(x))|^2]} \approx 0.127$.

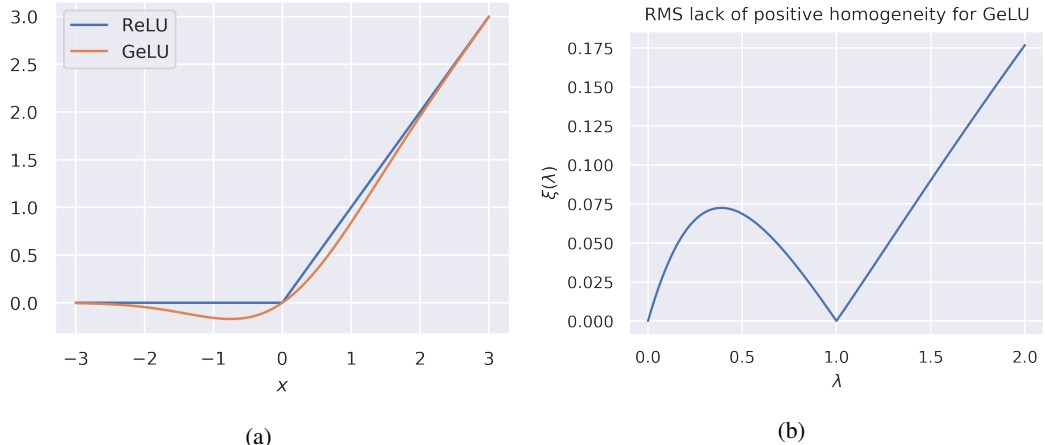

Figure 17: **(a)** The ReLU and GeLU functions. **(b)** Root-mean-square lack of positive homogeneity for the GeLU function, estimated using $10^5$ samples of $X$.

### E.3 Proof of theorem 3.3

*Proof.* The explicit description of $G(\mathrm{ReLU}, n)$ in table 1 is enough to show $G(\mathrm{ReLU}, n)$ *stabilizes* $\{\mathbb{R}_{\geq 0} e_i | i = 1, \ldots, n\}$. Indeed, any $A \in G(\mathrm{ReLU}, n)$ may be written as $PD$ where $D = \mathrm{diag}(a_i)$ for some $a_i > 0$ and $P$ is a permutation matrix associated to a permutation $\pi$. It suffices to show that $P$ and $\mathrm{diag}(a_i)$ each preserves $\{\mathbb{R}_{\geq 0} e_i | i = 1, \ldots, n\}$. First,

$$\mathrm{diag}(a_i)\{\mathbb{R}_{\geq 0} e_i | i = 1, \ldots, n\} = \{\mathbb{R}_{\geq 0} a_i e_i | i = 1, \ldots, n\} = \{\mathbb{R}_{\geq 0} e_i | i = 1, \ldots, n\}$$

since the $a_i > 0$ so scaling by $a_i$ preserves the ray $\mathbb{R}_{\geq 0} e_i$. Second,

$$P\{\mathbb{R}_{\geq 0} e_i | i = 1, \ldots, n\} = \{\mathbb{R}_{\geq 0} P e_i | i = 1, \ldots, n\} = \{\mathbb{R}_{\geq 0} e_{\pi(i)} | i = 1, \ldots, n\} = \{\mathbb{R}_{\geq 0} e_i | i = 1, \ldots, n\}$$

(the last equality is due to the fact that we only consider the set of rays, not the ordered tuple of rays).

Conversely, if $A \in GL(n)$ stabilizes $\{\mathbb{R}_{\geq 0} e_i | i = 1, \ldots, n\}$, then in particular for each $j$ $A e_j \in \mathbb{R}_{\geq 0} e_i$ for some $i$ and as $A$ is invertible, setting $j = \pi(i)$ yields a permutation of $\{1, \ldots, n\}$. Moreover, since $A e_j \neq 0$ ($A$ is invertible) there must be some $a_j > 0$ so that $A e_j = a_j e_i$. One can now verify that $A = PD$ where $P$ is the permutation matrix associated to $\pi$ and $D = \mathrm{diag}(a_j)$ which matches the description of $G(\mathrm{ReLU}, n)$ from table 1.

For the "moreover," we prove the contrapositive, namely that if $v = (v_1, \ldots, v_n) \in \mathbb{R}^n$ has at least 2 non-0 coordinates $v_i, v_j$, then the $G_{\mathrm{ReLU}}$-orbit of the ray $\mathbb{R}_{\geq 0} v$ that $v$ generates cannot be finite. Indeed, suppose $t \geq 0$ and let $D = \mathrm{diag}(1, \ldots, 1, t, 1, \ldots, 1)$ ($t$ in the $i$th position). The $n \times 2$ matrix $(v | Dv)$ has a $2 \times 2$ minor

$$\begin{pmatrix} v_i & t v_i \\ v_j & v_j \end{pmatrix} \text{ with deteriminant } (1 - t)v_i v_j \neq 0 \text{ as long as } t \neq 1. \tag{E.24}$$

Thus $v$ and $Dv$ are linearly independent, and hence define distinct rays, for all $t \neq 1$. It follows that any set of rays stabilized by $G(\mathrm{ReLU}, n)$ that contains $\mathbb{R}_{\geq 0} v$ is uncountable. $\square$

### E.4 Proof of proposition 3.4

To give a rigorous proof we use induction on the depth $k$; since this to some extent obscures the main point, we briefly outline an informal proof: consider a composition of 2 layers of the network $f$ with weights $W'$:

$$\sigma(A_{i+1} W_{i+1} \phi_\sigma(A_i)^{-1} \sigma(A_i W_i \phi_\sigma(A_{i-1})^{-1} x + A_i b_i) + A_{i+1} b_{i+1}). \tag{E.25}$$

Using the defining properties of $G_\sigma$ and $\phi_\sigma$, we can extract $A_i$ like

$$\sigma(A_i W_i \phi_\sigma(A_{i-1})^{-1} x + A_i b_i) = \phi_\sigma(A_i)\sigma(W_i \phi_\sigma(A_{i-1})^{-1} x + b_i). \tag{E.26}$$

The resulting copy of $\phi_\sigma(A_i)$ on the right hand side of eq. (E.26) is cancelled by the copy of $\phi_\sigma(A_i)^{-1}$ right-multiplying $W_{i+1}$ in eq. (E.25), so that eq. (E.25) reduces to

$$\sigma(A_{i+1}W_{i+1}\sigma(W_i\phi_\sigma(A_{i-1})^{-1}x + b_i) + A_{i+1}b_{i+1}).$$

In this way, in between any two layers $\ell_{i+1} \circ \ell_i$ the $A_i$ in $A_iW_i\phi_\sigma(A_{i-1})^{-1}$ and the $\phi_\sigma(A_i)^{-1}$ in $A_{i+1}W_{i+1}\phi_\sigma(A_i)^{-1}$ *cancel*. However, the factors $\phi_\sigma(A_m)$ and $\phi_\sigma(A_m)^{-1}$ appear on endpoints of the truncated networks $f_{\leq m}, f_{>m}$ and so they are not cancelled.

To keep track of $W, W'$ while using the notation from section 3, we write

$$\ell_i(x, W) = \sigma(W_ix + b_i) \text{ and } \ell_i(x, W') = \sigma(W'_ix + b'_i) \text{ for } i < k$$

and so on.

*Proof.* By induction on $k$, the depth of the network. The case $k = 1$ is trivial, since there $W' = W$ and there is nothing to prove. For $k > 1$ we consider 2 cases:

**Case** $m = 1$: In this case let $V = (W_2, b_2, \ldots, W_k, b_k)$ and $V' = (A_2W_2, A_2b_2, A_3W_3\phi_\sigma(A_2)^{-1}, A_3b_3, \ldots, W_k\phi_\sigma(A_{k-1}^{-1}), b_k)$, and let

$$g(x, V) = \ell_{k-1}(x, V) \circ \cdots \circ \ell_1(x, V) \text{ and } g(x, V') = \ell_{k-1}(x, V') \circ \cdots \circ \ell_1(x, V')$$

where $\ell_i(x, V) = \sigma(W_{i+1}x + b_{i+1})$ and similarly for $V'$. In other words, the weights $V, V'$ and function $g$ represent the architecture obtained by removing the earliest layer of $f$. Then, $f_{\leq 1}(x, W) = \sigma(W_1x + b_1)$ and on the other hand

$$f_{\leq 1}(x, W') = \sigma(A_1W_1x + A_1b_1) = \sigma(A_1(W_1x + b_1)).$$

Using the identity $\sigma(A_1z) = \phi_\sigma(A_1)\sigma(z)$ for any $z \in \mathbb{R}^{n_1}$, we obtain

$$f_{\leq 1}(x, W') = \phi_\sigma(A_1)\sigma(W_1x + b_1) = \phi_\sigma(A_1)\sigma(W_1x + b_1).$$

This shows $f_{\leq 1}(x, W') = \phi_\sigma(A_1) \circ f_{\leq 1}(x, W)$. Next, $f_{>1}(x, W) = g(x, V)$ but *because* $V'_1 = A_2W_2$ whereas $W'_2 = A_2W_2\phi_\sigma(A_1)^{-1}$

$$f_{>1}(x, W') = g(x, V') \circ \phi_\sigma(A_1)^{-1}.$$

By induction on $k$, we may assume $g(x, V) = g(x, V')$ and it follows that $f_{>1}(x, W') = f_{>1}(x, W) \circ \phi_\sigma(A_1)^{-1}$

**Case** $m > 1$: Defining $V, V'$ and $g$ as above, we observe that

$$\begin{aligned} f_{\leq i}(x, W) &= g_{\leq i-1}(\sigma(W_1x + b_1), V) \text{ and} \\ f_{\leq i}(x, W') &= g_{\leq i-1}(\phi_\sigma(A_1)^{-1}\sigma(A_1W_1x + A_1b_1), V') = g_{\leq i-1}(\sigma(W_1x + b_1), V'). \end{aligned} \tag{E.27}$$

By induction on $k$ we may assume $g_{\leq i-1}(x, V') = \phi_\sigma(A_i)g_{\leq i-1}(x, V)$ and so

$$f_{\leq i}(x, W') = \phi_\sigma(A_i)g_{\leq i-1}(\sigma(W_1x + b_1), V) = \phi_\sigma(A_i)f_{\leq i}(x, W).$$

Finally, $f_{>i}(x, W) = g_{>i-1}(x, V)$ and $f_{>i}(x, W') = g_{>i-1}(x, V')$ and we may assume by induction on $k$ that $g_{>i-1}(x, V') = g_{>i-1}(x, V) \circ \phi_\sigma(A_i)^{-1}$, hence $f_{>i}(x, W') = f_{>i}(x, W) \circ \phi_\sigma(A_i)^{-1}$. $\square$

## E.5 Proof of theorem 4.2

*Proof.* Observe that by proposition 3.4, $\tilde{f}_{>l} = f_{>l} \circ \phi_\sigma(A_l^{-1})$. Hence

$$S(f, \tilde{f}, l, \varphi) = \tilde{f}_{>l} \circ \varphi \circ f_{\leq i} = f_{>l} \circ \phi_\sigma(A_l^{-1}) \circ \varphi \circ f_{\leq l}.$$

If $\mathcal{S}$ contains $\phi_\sigma(G_{\sigma_{n_l}})$ we may choose $\varphi = \phi_\sigma(A_l)$ to achieve $S(f, \tilde{f}, l, \varphi) = f$ as functions. Similarly, $\tilde{f}_{\leq l} = \phi_\sigma(A_l) \circ f_{\leq l}$ so if $\mathcal{S}$ contains $\phi_\sigma(G_{\sigma_{n_l}})$ we may choose $\varphi = \phi_\sigma(A_l^{-1})$ to achieve $S(f, \tilde{f}, l, \varphi) = \tilde{f}$ as functions. In either case eq. (4.1) holds. $\square$

## E.6   Symmetries of the loss landscape

Given that the intertwiner group describes a large set of symmetries of a network, it is not surprising that it also provides a way of understanding the relationship between equivalent networks. Proposition 3.4 has an interpretation in terms of the loss landscape of model architecture. For any layer $n_i$ in $f$, the action of $G_{\sigma_{n_i}}$ on the weight space $\mathcal{W}$, translates to the obvious group action on the loss landscape.

**Corollary E.28.** *For any $1 \leq i \leq k$, the group $G_{\sigma_{n_i}}$ acts on $\mathcal{W}$ and for any test set $D_t \subset X \times Y$, model loss on $D_t$ is invariant with respect to this action. More precisely, if $\ell(\Phi(W), D)$ is the loss of $\Phi(W)$ on test set $D_t$, then for any $g \in G_{\sigma_{n_i}}$, $\ell(\Phi(W), D) = \ell(\Phi(gW), D)$.*

## E.7   Comparing capacities of stitching layers via discretization

Let $\mathrm{Mat}_{n \times n}(\mathbb{R})$ denote the space of $n \times n$ matrices. For $r = 1, \ldots, n$ let $\mathrm{Mat}^r_{n \times n}(\mathbb{R}) \subseteq \mathrm{Mat}_{n \times n}(\mathbb{R})$ denote the rank $r$ matrices, and let $G_{\mathrm{ReLU}_n}$ be as described in table 1. Suppose that each real dimension of $\mathrm{Mat}_{n \times n}(\mathbb{R})$ is replaced by a discrete grid $N(M, \epsilon) = \{-M + i\epsilon \mid i = 0, \ldots, \lfloor \frac{2M}{\epsilon} \rfloor - 1\}$ — here $\epsilon$ could represent the limits of numerical precision in a floating point number system, and $M$ could represent the maximum numerical magnitude. The size of each such grid is $\lfloor \frac{2M}{\epsilon} \rfloor$, and so the number of points in the resulting mesh grid $N(M, \epsilon)^{n^2} \subset \mathrm{Mat}_{n \times n}(\mathbb{R})$ is $\lfloor \frac{2M}{\epsilon} \rfloor^{n^2}$. We now estimate the number of points of $G_{\mathrm{ReLU}_n}$ and $\mathrm{Mat}^r_{n \times n}(\mathbb{R})$ in such a mesh grid.

$G_{\mathrm{ReLU}_n}$ is a disjoint union of $n!$ irreducible components, corresponding to the $n!$ possible permutations $P$ in table 1. Each of these components is $n$-dimensional, corresponding to the fact that the factor $D$ in table 1 is an arbitrary positive diagonal matrix. Hence we obtain

$$|G_{\mathrm{ReLU}_n} \cap N(M, \epsilon)^{n^2}| \approx n! \cdot (\frac{M}{\epsilon})^n. \tag{E.29}$$

On the other hand, any matrix $A \in \mathrm{Mat}^r_{n \times n}(\mathbb{R})$ can be written as $A = UV$ where $U \in \mathrm{Mat}_{n \times r}(\mathbb{R})$ and $V \in \mathrm{Mat}_{r \times n}(\mathbb{R})$. These $U$ and $V$ are not unique: given any invertible $r \times r$ matrix $W \in GL_r(\mathbb{R})$, we have $A = (UW)(W^{-1}V)$. From this we obtain the approximation[10]

$$|\mathrm{Mat}^r_{n \times n}(\mathbb{R}) \cap N(M, \epsilon)^{n^2}| \approx \frac{|\mathrm{Mat}_{n \times r}(\mathbb{R}) \cap N(M, \epsilon)^{nr}| \cdot |\mathrm{Mat}_{r \times n}(\mathbb{R}) \cap N(M, \epsilon)^{rn}|}{|GL_r(\mathbb{R}) \cap N(M, \epsilon)^{r^2}|} \tag{E.30}$$

$$\approx \frac{(\frac{2M}{\epsilon})^{nr}(\frac{2M}{\epsilon})^{rn}}{(\frac{2M}{\epsilon})^{r^2}} \tag{E.31}$$

$$\approx (\frac{2M}{\epsilon})^{2nr - r^2}. \tag{E.32}$$

It follows that

$$\log|G_{\mathrm{ReLU}_n} \cap N(M, \epsilon)^{n^2}| - \log|\mathrm{Mat}^r_{n \times n}(\mathbb{R}) \cap N(M, \epsilon)^{n^2}| \tag{E.33}$$

$$= \log(n!) + n\log(\frac{M}{\epsilon}) - (2nr - r^2)\log(\frac{M}{\epsilon} + \log 2). \tag{E.34}$$

Ignoring the term $(2nr - r^2)\log 2$, which is independent of $M, \epsilon$, we get the approximation

$$\log(n!) + n\log(\frac{M}{\epsilon}) - (2nr - r^2)\log(\frac{M}{\epsilon} + \log 2) \approx \log(n!) - ((2r-1)n - r^2)\log(\frac{M}{\epsilon}). \tag{E.35}$$

Next, we make the coarse approximation

$$\log(n!) = \sum_{k=1}^{n} \log k \approx \int_1^n \log x \, dx = n\log n - n; \tag{E.36}$$

---

[10]Here we ignore a significant subtlety: whether or not the multiplication map $\mathrm{Mat}_{n \times r}(\mathbb{R}) \times \mathrm{Mat}_{r \times n}(\mathbb{R}) \to \mathrm{Mat}_{n \times n}(\mathbb{R})$ induces a map $N(M, \epsilon)^{nr} \times N(M, \epsilon)^{rn} \to N(M, \epsilon)^{n^2}$ (with our naive setup it probably doesn't) and moreover whether the fibers of this map, which in the non-discretized case are generically isomorphic to $GL_r(\mathbb{R})$, have intersection with $N(M, \epsilon)^{nr} \times N(M, \epsilon)^{rn}$ of the expected size. We do not expect that these technical details will impact the takeaway of this analysis.

with this approximation the expression of eq. (E.35) is approximated as

$$\log(n!) - ((2r - 1)n - r^2)\log(\frac{M}{\epsilon}) \approx n\log n - n - ((2r - 1)n - r^2)\log(\frac{M}{\epsilon}). \quad \text{(E.37)}$$

From this we conclude that as long as

1. $r \geq 1$ (we actually already assumed this when defining $\mathrm{Mat}^r_{n \times n}(\mathbb{R})$) and

2. $\frac{M}{\epsilon} \gg n$, which roughly says that the number of grid points per dimension is greater than the number of rows (equivalently columns) in $\mathrm{Mat}_{n \times n}(\mathbb{R})$,

$$n\log n - n - ((2r - 1)n - r^2)\log(\frac{M}{\epsilon}) \leq n\log n - n - (n - 1)\log(\frac{M}{\epsilon}) \text{ using item 1} \quad \text{(E.38)}$$

$$= (n - 1)(\log n - \log(\frac{M}{\epsilon})) + \log n - n \quad \text{(E.39)}$$

$$< (n - 1)(\log n - \log(\frac{M}{\epsilon})) \text{ for } n > 1 \quad \text{(E.40)}$$

$$< 0 \text{ using item 2.} \quad \text{(E.41)}$$

The upshot is that our approximations and items 1 and 2 imply

$$\log|G_{\mathrm{ReLU}_n} \cap N(M, \epsilon)^{n^2}| - \log|\mathrm{Mat}^r_{n \times n}(\mathbb{R}) \cap N(M, \epsilon)^{n^2}| < 0, \text{ and hence} \quad \text{(E.42)}$$

$$|G_{\mathrm{ReLU}_n} \cap N(M, \epsilon)^{n^2}| < |\mathrm{Mat}^r_{n \times n}(\mathbb{R}) \cap N(M, \epsilon)^{n^2}|. \quad \text{(E.43)}$$

### E.8  Calculations related to dissimilarity measures (for section 5)

*Proof of lemma 5.5.* We must show that for any $x_1, \ldots, x_r \in \mathbb{R}^d$ and any $c_1, \ldots, c_r \in \mathbb{R}$ that

$$\sum_{i,j} c_i c_j \max(x_i \odot x_j) \geq 0.$$

We use the elementary "max $\geq$ mean" inequality:

$$\max(x_i \odot x_j) \geq \frac{1}{d}\sum_k x_{ik} x_{jk} = \frac{1}{d} x_i \cdot x_j$$

where on the right hand side "·" denotes dot product. This implies

$$\sum_{i,j} c_i c_j \max(x_i \odot x_j) \geq \frac{1}{d}\sum_{i,j} c_i c_j (x_i \cdot x_j) \geq 0$$

where the last inequality follows from the fact that the dot product is a kernel function. $\qquad\square$

The last string of inequalities also shows $\sum_{i,j} c_i c_j \max(x_i \odot x_j) > 0$ when $x_1, \ldots, x_r$ are linearly independent, but does not directly imply the converse. It would be interesting to know conditions on $x_1, \ldots, x_r$ that imply positive *definiteness* of the matrix $(\max(x_i \odot x_j))$.

*Proof of lemma 5.1.* By table 1 any $A \in G_{\mathrm{ReLU}}$ can be factored as $A = PD$ with $P$ a permutation matrix and $D$ a positive diagonal matrix, and we can obtain a similar factorization $B = QE$. Then

$$\mu(XA, YB) = \mu(XPD, YQE) = \mu(XP, YQ) = \mu(X, Y)$$

where the second equality uses the hypothesis that $\mu$ is invariant to right multiplication by positive diagonal matrices, and the third uses the hypothesis that $\mu$ is invariant to right multiplication by permutation matrices. $\qquad\square$

**Lemma E.44.** *Suppose $A$ is a matrix such that $\max(Ax_1 \odot Ax_2) = \max(Ax_1 \odot Ax_2)$ for all $x_1, x_2 \in \mathbb{R}^d$. Then, $A$ is of the form $PD$ where $P$ is a permutation matrix and $D$ is diagonal with diagonal entries in $\{\pm 1\}$.*

*Proof.* We only need the special case where $x = y$: observe that

$$\max(x \odot x) = \max\{x_1^2, \ldots, x_d^2\} = (\max\{|x_1|, \ldots, |x_d|\})^2 = |x|_\infty^2$$

This means that if $\max(Ax_1 \odot Ax_2) = \max(Ax_1 \odot Ax_2)$, then $A$ preserves the $\ell_\infty$ norm on $\mathbb{R}^d$, hence in particular preserves the unit hypercube in $\mathbb{R}^d$, and it is known that symmetries of the hypercube have the form $PD$ where $P$ is a permutation matrix and $D$ is diagonal with diagonal entries in $\{\pm 1\}$ (see for example [Ser77, §5.9]). $\qquad\square$

### E.9 Intertwiners and more general architecture features (justification of remark 3.6)

Here we briefly discuss how ubiquitous architecture features like batch normalization and residual connections interact with intertwiner groups. For simplicity in this section we only consider $\sigma = \text{ReLU}$.

#### E.9.1 Batch normalization

A batch normalization layer that takes as input $X \in \mathbb{R}^{b \cdot n}$ (where $b$ is the batch size and $n$ is the dimension of the layer) and returns

$$\tilde{X} \operatorname{diag}(\tilde{X}^T \tilde{X})^{-1} \operatorname{diag}(\gamma) + \beta \text{ where } \tilde{X} = X - \mathbf{1}\mathbf{1}^T X$$

and where $\gamma, \beta \in \mathbb{R}^n$ are the "gain" and "bias" parameters of the batch normalization layer, is *invariant* under independent scaling of coordinates, that is transformations of the form $X \leftarrow XD$ where $D$ is an $n \times n$ positive diagonal matrix (see e.g. [BMC15]). Hence a $k$-layer ReLU MLP as in section 3 enhanced with batch normalization (*pre*-activation, as is standard) is invariant under the action of the slightly larger group $\prod_{l=1}^{k-1}(\mathbb{R}_{>0}^{n_l} \rtimes G_{\text{ReLU}_{n_l}})$, where the action is given by[11]

$$\begin{aligned}
&(c_1, A_1, \ldots, c_{k-1}, A_{k-1}) \cdot (W_1, \gamma_1, \beta_1, \ldots, W_{k-1}, \gamma_{k-1}, \beta_{k-1}, W_k, b_k) \\
&= (A_1 W_1, \pi(A_1) \operatorname{diag} c_1 \gamma_1, \pi(A_1) \operatorname{diag} c_1 \beta_1, \\
&A_2 W_2 (\pi(A_1) \operatorname{diag} c_1)^{-1}, \pi(A_2) \operatorname{diag} c_2 \gamma_2, \pi(A_2) \operatorname{diag} c_2 \beta_2, \\
&\ldots, A_{k-1} W_{k-1} (\pi(A_{k-2}) \operatorname{diag} c_{k-2})^{-1}, \pi(A_{k-1}) \operatorname{diag} c_{k-1} \gamma_{k-1}, \pi(A_{k-1}) \operatorname{diag} c_{k-1} \beta_{k-1}, \\
&W_k (\pi(A_{k-1}) \operatorname{diag} c_{k-1})^{-1}, b_k).
\end{aligned} \tag{E.45}$$

Here,

- $c_l \in \mathbb{R}_{>0}^{n_l}$ and $A_l \in G_{\text{ReLU}_{n_l}}$, for all $l = 1, \ldots, k-1$
- $\pi : G_{\text{ReLU}_{n_l}} \to \Sigma_{n_l}$ is the homomorphism setting the positive entries to 1.

The key point is that we get another factor of $\mathbb{R}_{>0}^{n_l}$ at each layer. We also note that the space of matrices of the form $\pi(A_l) \operatorname{diag} c_l$ is, incidentally, exactly $G_{\text{ReLU}_{n_l}}$, and that using eq. (E.45) one can generalize proposition 3.4 and theorem 4.2 to the case of networks with batch normalization.

#### E.9.2 Residual connections

We expand on remark 3.6 and explain what exactly transpires with residual connections below.

Suppose we have a $k$-layer MLP as in section 3 (again for simplicity with $\sigma = \text{ReLU}$), together with residual connections between a set of layers $R = \{r_1, \ldots, r_m\} \subseteq \{2, \cdots, k-1\}$[12]:

$$\mathbb{R}^{n_0} \xrightarrow{\sigma W_1} \cdots \xrightarrow{\sigma W_{r_i-1}} \mathbb{R}^{n_{r_i-1}} \underbrace{\xrightarrow{\sigma W_{r_{i-1}+1}} \cdots \xrightarrow{W_{r_i}}}_{\text{id}} \mathbb{R}^{n_{r_i}} \xrightarrow{\sigma} \cdots \mathbb{R}^{n_{k-1}} \xrightarrow{W_k} \mathbb{R}^{n_{L+1}} \tag{E.46}$$

(for legibility biases $b_l$ are suppressed). In addition we assume that the depth of each residual block is some fixed, that is $r_i - r_{i-1} = b = \text{constant}$ for all $i$.

First, we claim that a $(A_1, \ldots A_{k-1}) \in \prod_{l=1}^{L} G_{\text{ReLU}_{n_l}}$ stabilizes the function $f$ *if* (not claiming if and only if) $A_{r_i} = A_{r_j}$ for all $r_i, r_j \in R$. To see this suppose $g_i(x, W)$, $i = 1, \ldots, m$ are the depth $b$ feedforward networks of the residual blocks, so that the $f_{\leq r_i}$ is given by

$$\begin{aligned}
f_{\leq r_i}(x, W) &= f_{\leq r_{i-1}}(x, W) + g_i(f_{\leq r_{i-1}}(x, W), W) \text{ where} \\
g_i(z, W) &= \sigma(W_{r_i+b} \sigma(\cdots \sigma(W_{r_{i-1}+1} z) \cdots)).
\end{aligned} \tag{E.47}$$

---

[11]As is best practice we omit the biases on the linear layers $\ell_l$ for $l < k$, since they would be redundant now that we have biases on batch norm layers.

[12]In particular we assume there is at least one linear layer before the first outgoing/after the last incoming residual connection, as occurs in e.g. ResNets.

Assuming by induction on $i$ that proposition 3.4 applies *at the residual connections in $R$* we note that with weights $W'$ eq. (E.47) turns into

$$\begin{aligned}
f_{\leq r_i}(x, W') &= f_{\leq r_{i-1}}(x, W') + g_i(f_{\leq r_{i-1}}(x, W'), W') \\
&= A_{r_{i-1}} f_{\leq r_{i-1}}(x, W) + g_i(A_{r_{i-1}} f_{\leq r_{i-1}}(x, W), W').
\end{aligned} \tag{E.48}$$

Note that proposition 3.4 applies directly to the $g_i$, so we may compute $g_i(z, W') = A_{r_i} g_i(A_{r_{i-1}}^{-1} z, W)$. Hence

$$\begin{aligned}
f_{\leq r_i}(x, W') &= A_{r_{i-1}} f_{\leq r_{i-1}}(x, W) + A_{r_i} g_i(A_{r_{i-1}}^{-1} A_{r_{i-1}} f_{\leq r_{i-1}}(x, W), W) \\
&= A_{r_{i-1}} f_{\leq r_{i-1}}(x, W) + A_{r_i} g_i(f_{\leq r_{i-1}}(x, W), W).
\end{aligned} \tag{E.49}$$

and we see that the only way $f_{\leq r_i}(x, W') = B f_{\leq r_i}(x, W)$ for some matrix $B$ is if $A_{r_{i-1}} = A_{r_i}$, proving our claim. This also shows that if $A_r$ denotes the common value of the $A_{r_i}$ for $r_i \in R$, we have $f_{\leq r_i}(x, W') = A_r f_{\leq r_i}(x, W)$ for all $i$. It is also true that $f_{>r_i}(x, W') = f_{>r_i}(A_r^{-1} x, W)$: observe that

$$f_{>r_i}(x, W) = f_{>r_{i+1}}(x + g_{i+1}(x, W), W). \tag{E.50}$$

By *descending* induction on $k$, we may assume $f_{r_{i+1}}(x, W') = f_{r_{i+1}}(A_r^{-1} x, W)$, and as above $g_{i+1}(z, W') = A_r g_{i+1}(A_r^{-1} z, W)$, so that with weights $W'$ eq. (E.50) becomes

$$\begin{aligned}
f_{>r_i}(x, W') &= f_{>r_{i+1}}(x + g_{i+1}(x, W'), W') \\
&= f_{>r_{i+1}}(A_r^{-1}(x + A_r g_{i+1}(A_r^{-1} x, W)), W) \\
&= f_{>r_{i+1}}(A_r^{-1} x + g_{i+1}(A_r^{-1} x, W), W) = f_{>r_i}(A_r^{-1} x, W).
\end{aligned} \tag{E.51}$$

as claimed.

Finally, we describe how stitching fails inside a residual block. Suppose we use weights $W_l$ for $l \leq r_i + j$ where $0 < j < b$) (recall $b = $ depth of our basic block) and weights $W'_l$ for $l > r_i + j$. The resulting stitched network is (attempting to use indentation to increase legibility)

$$\begin{aligned}
f_{r_{i+1}}(& \\
&f_{\leq r_i}(x, W) + g_{>j}( \\
&\qquad\qquad \varphi g_{\leq j}(f_{\leq r_i}(x, W), W), \\
&\qquad\qquad W'), \\
&W').
\end{aligned} \tag{E.52}$$

By proposition 3.4 $g_{>j}(z, W') = A_r g_{>j}(A_{r_i+j}^{-1} z, W)$, and we have shown $f_{>r_{i+1}}(z, W') = f_{>r_{i+1}}(A_r^{-1} z, W)$. Combining these facts eq. (E.52) becomes

$$\begin{aligned}
f_{r_{i+1}}(& \\
&A_r^{-1} f_{\leq r_i}(x, W) + A_r^{-1} A_r g_{>j}( \\
&\qquad\qquad A_{r_i+j}^{-1} \varphi g_{\leq j}(f_{\leq r_i}(x, W), W), \\
&\qquad\qquad W), \\
&W).
\end{aligned} \tag{E.53}$$

*Even after cancelling to remove the $A_r^{-1} A_r$ and in the ideal case where $\varphi = A_{r_i+j}$, we are left with an extra factor of $A_r^{-1}$ left multiplying $f_{\leq r_i}(x, W)$:*

$$\begin{aligned}
f_{r_{i+1}}(& \\
&A_r^{-1} f_{\leq r_i}(x, W) + g_{>j}( \\
&\qquad\qquad g_{\leq j}(f_{\leq r_i}(x, W), W), \\
&\qquad\qquad W), \\
&W).
\end{aligned} \tag{E.54}$$

# F    Network dissection details

Here we include some supplementary results and experiments for examining coordinate basis interpretability with network dissection [Bau+17].

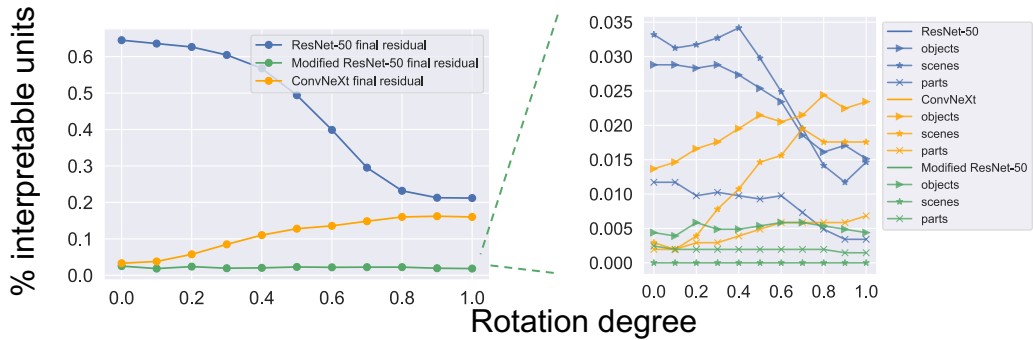

Figure 18: Supplement for the same network dissection experiment for the ResNet-50, modified ResNet-50, and ConvNeXt models in fig. 3, highlighting the categories of interpretable units for each model and basis on the right. The y-axis for the plot on the right is distinct concepts.

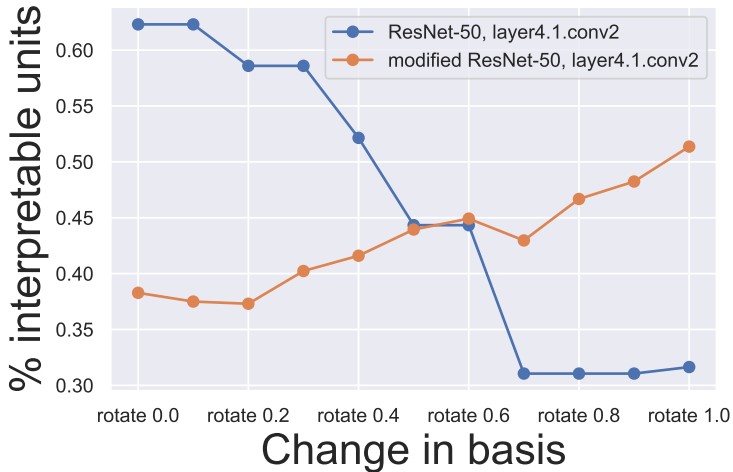

Figure 19: Fraction of network dissection interpretable units under rotations of the representation basis for a ResNet-50 and a modified ResNet-50 without an activation function on the residual output.

## F.1 Network dissection methodology

The Broden concept dataset, compiled by Bau et al., contains pixel-level annotations for hierarchical concepts including colors, textures, objects, and scenes. For every channel activation, network dissection assesses the binary segmentation performance with every visual concept from Broden. The method first computes the channel activation for every Broden image. The distribution of the activations for the channel is used to binarize the activation (where we threshold by the top $0.5\%$ of all activations for the channel) to define a segmentation mask for the channel activation which is interpolated to the size of the input image. If the Intersection over Union (IoU) of the activation segmentation mask and a concept mask is high enough (namely where IoU $> 0.04$), network dissection labels the activation an interpretable detector for the concept.

## F.2 Additional experiments

fig. 18 breaks down the categories of interpretable units for the models and rotations examined in fig. 3. The number of interpretable units tends to be dominated by the object and scene concept detectors for the ResNet-50 and the ConvNeXt models.

We also perform an analogous network dissection experiment to section 6 within the residual blocks for the normal and modified ResNet-50 in fig. 19. Like the ConvNeXt model in fig. 3 we find that, surprisingly, the percentage of interpretable units actually tends to increase away from the activation basis.

Per-concept breakdowns produced by network dissection for three different rotation powers in the experiment in fig. 3 are given for the ResNet-50 in fig. 20, the modified ResNet-50 (without a ReLU activation function on the residual output) in fig. 22, and the ConvNeXt in in fig. 24. We also include the units with the highest concept intersection over union scores for the the three representative rotations for the ResNet-50 in fig. 21, the modified ResNet-50 (without a ReLU activation function on the residual output) in fig. 23, and the ConvNeXt in in fig. 25

### F.3 Model training details

We train a ResNet-50 without ReLU (or any activation function) on the residual blocks in PyTorch using [Lec+22] on ImageNet [Den+09]. We train with SGD with momentum for 88 epochs with a cyclic learning rate rate of 1.7, label smoothing of 0.1, a batch size of 512, and weight decay of $10^{-4}$. The model achieves 76.1% top-1 accuracy. We use pretrained weights for the ResNet-50 (unmodified) and ConvNeXt models from [MR10] and [Wig19] respectively.

## G  Dataset Details

CIFAR-10: CIFAR-10 is covered by the MIT License (MIT). We use canonical train/test splits (imported using torchvision).

Broden: the code used to generate the dataset is covered by the MIT license.

ImageNet: ImageNet is covered by CC-BY 4.0. We use canonical train/test splits.

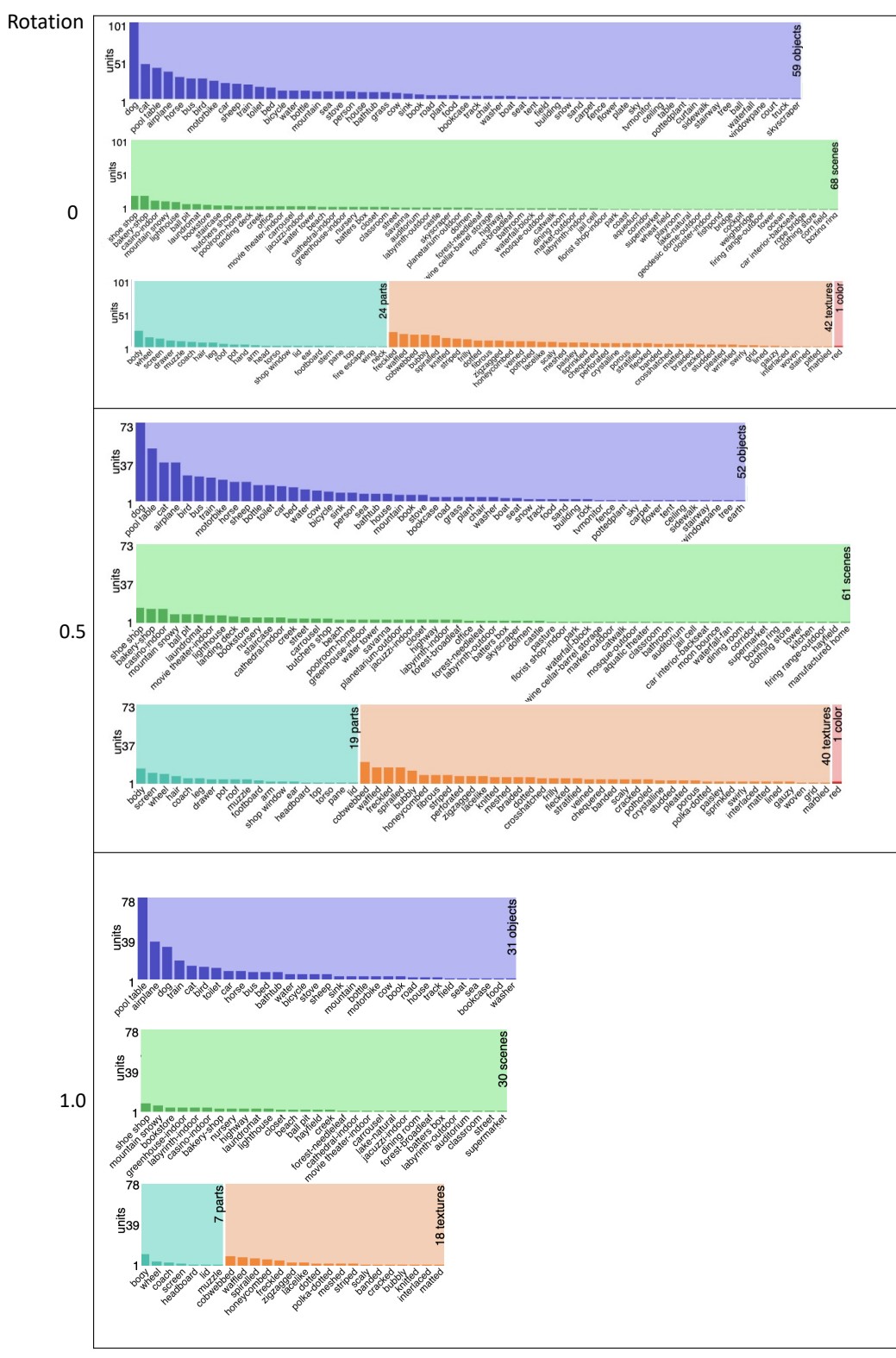

Figure 20: Network dissection bar graph of categories of unique concepts at three different rotation powers for the ResNet-50 model in fig. 3.

Rotation

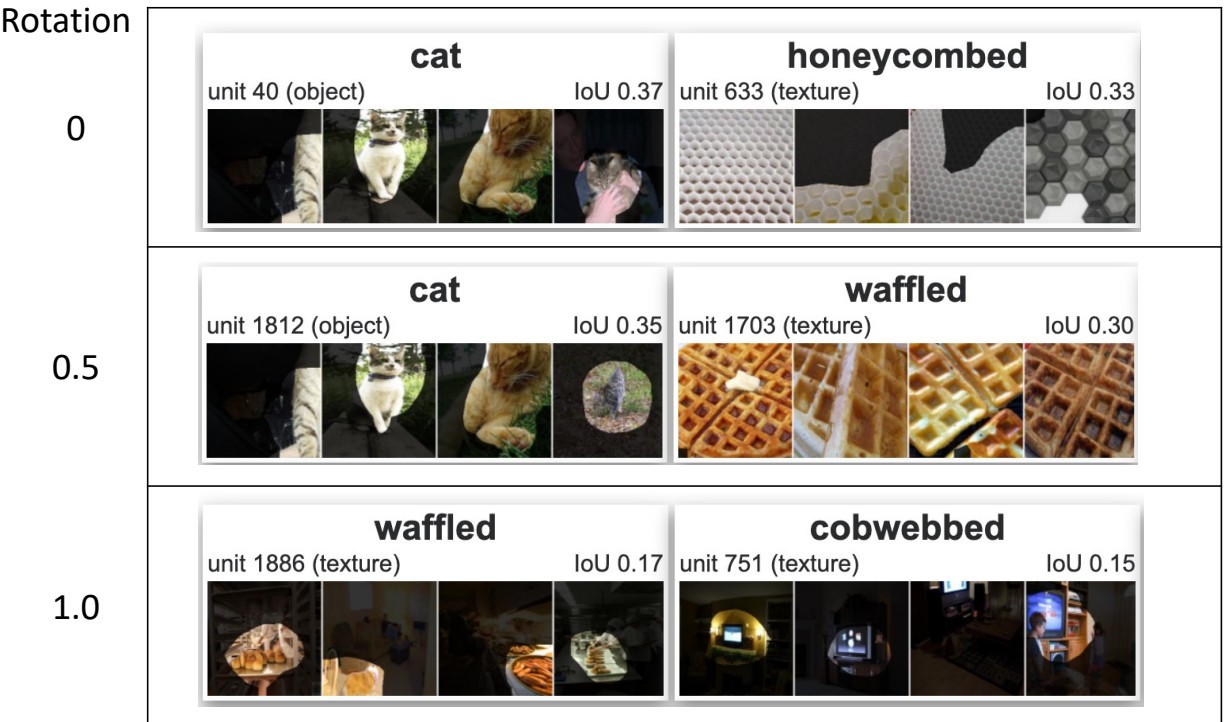

Figure 21: Top two highest scoring units for network dissection at three different rotation powers for the ResNet-50 model in fig. 3.

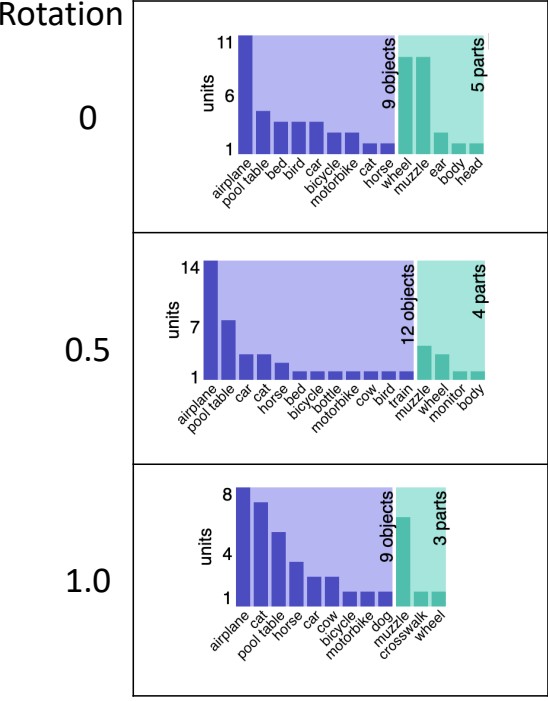

Figure 22: Network dissection bar graph of categories of unique concepts at three different rotation powers for the modified ResNet-50 model (without a ReLU activation function on the residual output) in fig. 3.

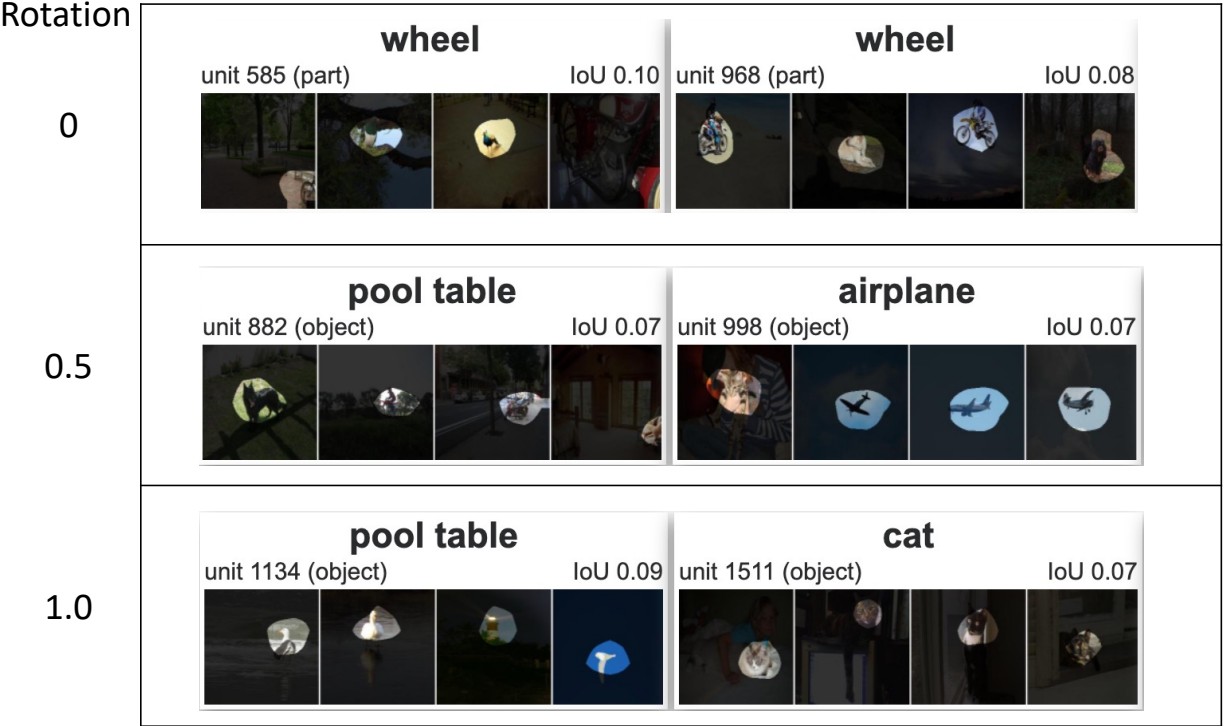

Figure 23: Top two highest scoring units for network dissection at three different rotation powers for the modified ResNet-50 model (without a ReLU activation function on the residual output) in fig. 3.

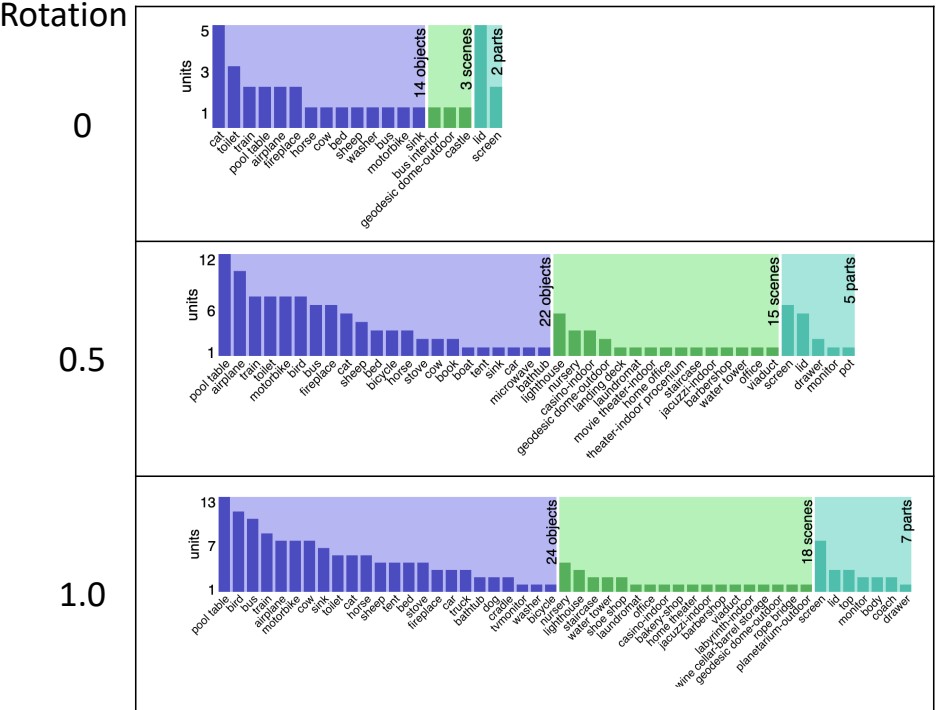

Figure 24: Network dissection bar graph of categories of unique concepts at three different rotation powers for the ConvNeXt model in fig. 3.

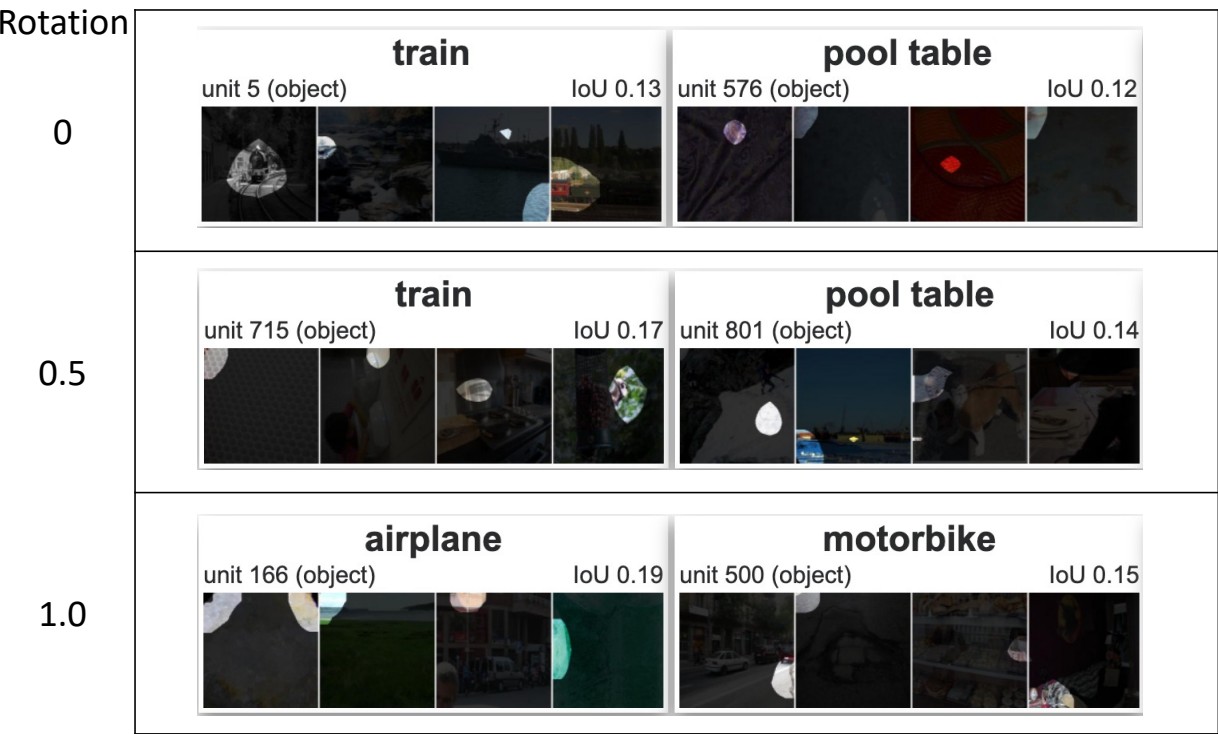

Figure 25: Top two highest scoring units for network dissection at three different rotation powers for the ConvNeXt model in fig. 3.

(a) Myrtle CNN architecture, summary courtesy of torchinfo.

(b) Myrtle CNN architecture without batch norm (only used in section 3.3), summary courtesy of torchinfo.

```
Layer (type:depth-idx)          Output Shape        Param #
========================================================================
ResNet                          --                  --
├─Conv2d: 1-1                   [1, 16, 32, 32]     432
├─BatchNorm2d: 1-2              [1, 16, 32, 32]     32
├─ReLU: 1-3                     [1, 16, 32, 32]     --
├─Sequential: 1-4              [1, 64, 8, 8]       --
│    └─BasicBlock: 2-1          [1, 16, 32, 32]     4,672
│    └─BasicBlock: 2-2          [1, 16, 32, 32]     4,672
│    └─BasicBlock: 2-3          [1, 16, 32, 32]     4,672
│    └─BasicBlock: 2-4          [1, 32, 16, 16]     13,952
│    └─BasicBlock: 2-5          [1, 32, 16, 16]     18,560
│    └─BasicBlock: 2-6          [1, 32, 16, 16]     18,560
│    └─BasicBlock: 2-7          [1, 64, 8, 8]       55,552
│    └─BasicBlock: 2-8          [1, 64, 8, 8]       73,984
│    └─BasicBlock: 2-9          [1, 64, 8, 8]       73,984
├─Linear: 1-5                   [1, 10]             650
========================================================================
Total params: 269,722
Trainable params: 269,722
Non-trainable params: 0
Total mult-adds (M): 40.55
========================================================================
Input size (MB): 0.01
Forward/backward pass size (MB): 3.01
Params size (MB): 1.08
Estimated Total Size (MB): 4.11
========================================================================
```

(a) Our ResNet20 architecture, summary courtesy of torchinfo.

```
Layer (type:depth-idx)          Output Shape        Param #
========================================================================
BasicBlock                      --                  --
├─Conv2d: 1-1                   [1, 16, 32, 32]     2,304
├─BatchNorm2d: 1-2              [1, 16, 32, 32]     32
├─ReLU: 1-3                     [1, 16, 32, 32]     --
├─Conv2d: 1-4                   [1, 16, 32, 32]     2,304
├─BatchNorm2d: 1-5              [1, 16, 32, 32]     32
├─Sequential: 1-6              [1, 16, 32, 32]     --
├─ReLU: 1-7                     [1, 16, 32, 32]     --
========================================================================
Total params: 4,672
Trainable params: 4,672
Non-trainable params: 0
Total mult-adds (M): 4.72
========================================================================
Input size (MB): 0.07
Forward/backward pass size (MB): 0.52
Params size (MB): 0.02
Estimated Total Size (MB): 0.61
========================================================================
```

(b) Internals of the 1st BasicBlock (the sequential contains the residual connection).

```
Layer (type:depth-idx)          Output Shape        Param #
========================================================================
ResNetImageNet                  --                  --
├─Conv2d: 1-1                   [1, 64, 32, 32]     1,728
├─BatchNorm2d: 1-2              [1, 64, 32, 32]     128
├─ReLU: 1-3                     [1, 64, 32, 32]     --
├─Sequential: 1-4              [1, 512, 4, 4]      --
│    └─BasicBlock: 2-1          [1, 64, 32, 32]     73,984
│    └─BasicBlock: 2-2          [1, 64, 32, 32]     73,984
│    └─BasicBlock: 2-3          [1, 128, 16, 16]    230,144
│    └─BasicBlock: 2-4          [1, 128, 16, 16]    295,424
│    └─BasicBlock: 2-5          [1, 256, 8, 8]      919,040
│    └─BasicBlock: 2-6          [1, 256, 8, 8]      1,180,672
│    └─BasicBlock: 2-7          [1, 512, 4, 4]      3,673,088
│    └─BasicBlock: 2-8          [1, 512, 4, 4]      4,720,640
├─Linear: 1-5                   [1, 10]             5,130
========================================================================
Total params: 11,173,962
Trainable params: 11,173,962
Non-trainable params: 0
Total mult-adds (M): 555.43
========================================================================
Input size (MB): 0.01
Forward/backward pass size (MB): 9.83
Params size (MB): 44.70
Estimated Total Size (MB): 54.54
========================================================================
```

(a) Our ResNet18 architecture, summary courtesy of torchinfo.

```
Layer (type:depth-idx)          Output Shape        Param #
========================================================================
BasicBlock                      --                  --
├─Conv2d: 1-1                   [1, 64, 32, 32]     36,864
├─BatchNorm2d: 1-2              [1, 64, 32, 32]     128
├─ReLU: 1-3                     [1, 64, 32, 32]     --
├─Conv2d: 1-4                   [1, 64, 32, 32]     36,864
├─BatchNorm2d: 1-5              [1, 64, 32, 32]     128
├─Sequential: 1-6              [1, 64, 32, 32]     --
├─ReLU: 1-7                     [1, 64, 32, 32]     --
========================================================================
Total params: 73,984
Trainable params: 73,984
Non-trainable params: 0
Total mult-adds (M): 75.50
========================================================================
Input size (MB): 0.26
Forward/backward pass size (MB): 2.10
Params size (MB): 0.30
Estimated Total Size (MB): 2.66
========================================================================
```

(b) Internals of the 1st BasicBlock (the sequential contains the residual connection).