# OpenReview forum: "On the Symmetries of Deep Learning Models and their Internal Representations"
_NeurIPS.cc/2022/Conference — NeurIPS 2022 Accept_

### Official Review · Reviewer_cBdu · 2022-07-09

**Rating:** 7
**Confidence:** 4
**Soundness:** 3 good
**Presentation:** 4 excellent
**Contribution:** 3 good

**Summary:**

This paper studies symmetries of the space of neural network parameters, i.e., invertible transformations of the parameters which leave the forward function invariant.  They compute this set of symmetries, called the intertwiner group, for networks with different activation functions.  As those authors note, these symmetries have been defined and studied before by other authors.  The primary focus of this work is on the impact of these symmetries on learned hidden representations in the network.  The author pursue a line of research which asks to what extent networks trained from different random initializations learn effectively the same hidden features?  In particular, what amount of variation is due to parameter space symmetry?  Through splicing experiments (Sec. 4), they shown that representations from one network can spliced into another using the intertwiner group for only a small drop in performance.  In Section 5, the authors compute directly using two novel metrics how close learned hidden representations are to being related by a single intertwiner for the whole dataset.  Section 6 shows, using network dissection, that the significance neuron activations (versus linear combinations of activations) to interpretability depends on the choice of activation function.

**Questions:**


## Questions
- Line 165, skip connections: I think this can be addressed by choosing a different group action than the one in Prop 3.4 which acts by the same A_i on layer which are tied by a skip connection.
- Why call G_{relu}-Procrustes distance a distance when it is larger for more similar inputs?  Perhaps "similarity measure"?
- Figure 5: What is meant by 5 runs of the experiment? Does this mean you trained 10 independently initialized networks and reported distances for 5 of the pairs?  What happens to the value is we compare all 5*4/2 pairs? The similarity scores don't seem that high, implying a high amount of variability, so it seems a bit surprising the numbers are so consistent between pairs.
- Figure 6: Grelu CKA seems to fall off faster with depth than CKA, why do you think this is?
- The argument for needing Grelus version of Procrustes and CKA (it seemed to me) was that the similarity measure should be adapted to the symmetry group of the network.  However, empirically they are very well correlated (esp. -Procrustes.)  If so, then doesn't that undermine the need for a new metric?  Also, while the gap between the rows is explained and makes sense, why are the rows in Figure 5 so well-correlated?
- Line 310-311: Since the rotation is random and the activations should be independent of basis, how do you explain the surprising increase in performance?


## Minor Points
- Figure 4 and Figure 5 are actually tables and should read ``Table''
- Figure 3 the y-axis is written as percent.  Is it percentage or fraction?
- Figure 3 the x-axis is again a fraction of the sampled rotation.  I wouldn't use the word degree which can be confused with rotations out of 360 degrees.
- Line 109 You should clarify $k$ is the last layer before giving a different definition for $l_k$.
- Line 173 should $i$ be $l$?
- In the example in Appendix C (second part), A is defined, but then the 2 and 1 switch places in the equation.  Is it correct?
- Line 206: "we are only allow"
- Line 254  " Grelu-orthogonal" should be "orthogonal-procrustes"?
- Line 260 $\tilde{X}$
- Line 306 in Figure 3

## Post-Response Edit
I appreciate the author's response to my questions.  I particularly appreciate the explanation on "Empirical correlation between G_{relu} and orthogonal dissimilarity measures" and alternate analysis of capacity or stitching maps.   The additional experiment examining "decay of G_{relu}," rules out one possible explanation, but I suppose it remains for future work to examine more closely.

 I looked at the MLP experiment in Appendix D.4 as an attempt to answer whether this hypothesis holds for MLPs as well as CNNs.  It seems the experiment is negative result in that symmetry does not well explain the difference between hidden reps here.  That's too bad, but I don't think it detracts overly from the current paper.  It does raise the question for future study of when parameter-space symmetry does or does not account for variation in latent representation.

I have read the other reviews and author responses, and I looked at the revised submission.   I will maintain my positive assessment.

**Limitations:**

Yes, the author include a clear limitations section.

**Strengths And Weaknesses:**

## Strengths

- This work presents a nice mixture of theoretical and empirical results connecting symmetries of parameter space to learned hidden features.  Although the authors are not the first to point out these symmetries, their treatment is rigorous, more general, and easy to follow.  Unlike earlier work, they connect parameter space symmetries to the question of variability of learned hidden representations.  The intertwiner symmetries may be considered change-of-basis symmetries for hidden layers.  If one is going to ask how much learned hidden features vary for different initializations, it is natural to ask how much of this variation is due to symmetry alone.  Network splicing where the splices are given by intertwiners is a clever way to get at this question.  The empirical splicing results in Section 4 and the computed similarity measures in Section 5 show that much (but not all) of the variability is reducible to symmetries of parameter space.
- The definition of intertwiners given (Defn 3.2) is more general than that given by other authors, allowing for the group action to be transformed after applying non-linearity.  This expands the set of a potential symmetries as in the examples in Figure 4.
- Splicing with G_{relu} requires optimizing over the (large and discreet) permutation group.  This would appear to be a difficult problem but the authors achieve good results with a convex relaxation to doubly stochastic matrices.
- I read parts of the appendix, which is quite thorough, provides good proofs for the statements in the paper, additional results and examples which are helpful for understanding.  I want to point out for the other reviewers there are some small mistake corrections and improvements to the main pointed out in the appendix (e.g. A --> A_m in (5)).

## Weaknesses
- The splicing comparison in Sec. 4 could be more exhaustive.  a) The comparison is done only for convolutional networks across the channels using 1x1 convolutions.  Does a similar result hold for MLPs?  b) Moreover, the justification that the rank 1, rank 2, and rank 4 1x1 have greater capacity to achieve lower error doesn't seem that convincing to me.  The advantage of Grelu is that it has access to a very large space of (full rank) permutation symmetries.  Since these are discreet, they don't count towards dimension, but they do give access to many useful transforms in GL_n.  In other words, dimension may not be the fairest way to compare.
- I personally find the interpretability argument put forward in Section 6 to be not that strong and somewhat aside from the main point of the paper.  The design of artificial neural networks using component-wise activations in order to model distinct neurons means that the distinct choice of basis in the activation vector space is baked into the network design.  The fact that linear combinations of neuron activations are less interpretable neuron activations is thus reasonable on its own without reference to the (also interesting) observation regarding the permutation symmetry of component-wise activations.
- I find it a bit hard to interpret the results in Section 5.  In particular, I lack a scale/baseline for, e.g., the numbers in Figure 6.  What does a Grelu-CKA of 0.63 for layer 6 of two models mean in terms of understanding how much of the variability of trained representations comes from symmetry?  Or are there other lessons I should draw from the results?

---

> ### Author Response · Authors · 2022-08-01
> **Capacity estimates, interpretation of dissimilarity measure values and variability/consistency**
>
> Thanks very much for your thorough read and valuable feedback. We will attempt to address your questions in turn.
>
> **Regarding capacity of restricted-rank matrices vs. $G_{\mathrm{ReLU}}$:** we agree that dimension is only one (possibly misleading) measure of capacity --- in particular rank itself is often a useful measure of capacity and there $G_{\mathrm{ReLU}}$ clearly wins. In case it's of interest, we've included in E.7 a slightly different analysis of capacity motivated by the theory of $ \epsilon $-nets. Its upshot is that if one works with very few floating point bits, $G_{\mathrm{ReLU}}$ has more elements since the factor of $ n! $ takes over, whereas with many floating point bits $B$ the factor of $2^B$ bits per dimension takes over and we get the same conclusion as when comparing dimensions.
>
> We have also included in sec. D.6 a small experiment stitching with full 1-by-1 convolutions with LASSO regularization -- varying the strength of the lasso penalty could be viewed as an alternative way of controlling capacity. We find that $G_{\mathrm{ReLU}}$ layers seem to achieve relatively low stitching penalties given their sparsity levels.
>
> **Interpretation of the results in Section 5:** it seems a somewhat difficult question to say what 0.63 means quantitatively! As a first pass, it's easier to say what happens at extremes: when $G_{\mathrm{ReLU}}$-CKA = 1, we know that the kernel matrices $K, L$ of Def. 5.3 are roughly parallel (roughly only since the HSIC isn't *exactly* the same as the inner product). This means that $\max (\tilde{x}\_{i} \odot \tilde{x}\_{j}) \propto \max (\tilde{y}\_{i} \odot \tilde{y}\_{j})$ with the *same constant of proportionality shared across all* $ i, j$. From here, we do not go so far as to claim that $\tilde{X}, \tilde{Y}$  differ by a permutation, although Lem. E.44 isn't far off. In the case where $G_{\mathrm{ReLU}}$-CKA = 0, the kernel matrices $K, L$ of Def. 5.3 are roughly orthogonal. This seems to have a less transparent interpretation. For intermediate values like 0.63, we still of course can extract a statement about cosine similarity of $K, L$ but it's less clear how this quantitatively measures symmetry. Understanding intermediate values seems to be a challenge with CKA measures generally, which is perhaps why they have mostly been used to make qualitative, relative comparisons (e.g. the features at layer $l$ are more related-by-symmetry than those at layer $m$).

---

> ### Author Response · Authors · 2022-08-01
> **Decay of $G_{\mathrm{ReLU}}$-CKA and empirical correlation between $G_{\mathrm{ReLU}}$ and orthogonal dissimilarity measures**
>
> Thanks very much for your thorough read and valuable feedback. We will attempt to address your questions in turn.
>
> **The decay of $G_{\mathrm{ReLU}}$-CKA:** we are very curious about this as well. One question we had has whether this could have something to do with dimensionality: all the networks we looked at had the feature that their channel dimension grows exponentially with depth (as seen in the last 3 figures of the appendix). In sec. D.8 we investigate this question with a new experiment using a CNN of constant channel dimension. The upshot is that it still exhibits decay of  $G_{\mathrm{ReLU}}$-CKA with depth, suggesting this phenomena is likely not an artifact of varying channel dimension. In D.8 we also include some further speculations on what else might be the cause of this phenomena.
>
> **Empirical correlation between $G_{\mathrm{ReLU}}$ and orthogonal dissimilarity measures:** this is indeed striking. One way of looking at this is that the intertwiner group accounts for a lot of the variation in hidden feature representations, as we also see in the stitching experiments, and as such it makes sense that the $G_{\mathrm{ReLU}}$ and orthogonal dissimilarity measures are correlated. As for *why* $G_{\mathrm{ReLU}}$ accounts for so much of the variation, we are tempted to propose a couple (not necessarily related) conjectures
> - Possibly this stems from density properties of the permutations in the orthogonal groups (either in the Riemannian metric on $O(d)$ or perhaps a matrix norm such as the Frobenius or nuclear norm). However, we were unable to find much existing work on this question (the closest we could find was http://www.math.lsa.umich.edu/~barvinok/ort.pdf, which considers a somewhat different notion of approximation), and based on a few initial attempts estimating this density seems non-trivial.
> - Another possibility is that this is related to the *non-negativity* of the hidden features we look at (all are outputs of $\mathrm{ReLU}$ layers): the only orthogonal matrices that preserve $\mathbb{R}\_{\geq 0}^n \subset \mathbb{R}^n$ are the permutations!

---

> ### Author Response · Authors · 2022-08-02
> **Interpretability (or lack thereof) of linear combinations of neurons, more exaustive experiments, etc.**
>
> Thanks very much for your thorough read and valuable feedback. We will attempt to address your questions in turn.
>
> **More exhaustive experiments**: we include more pair-wise comparisons for the stitching penalties in figure 1, 6, and 7, as well as the Procrustes/CKA experiments in Figures 2, 4, 9, 10, and 11 (all noted in the figure captions, as well). We also include a new experiment for the MLP layers of vision transformers on CIFAR-10 (D.4 in the appendix).
>
> **Interpretability (or lack thereof) of linear combinations of neurons:** while we have no argument with your line of reasoning, it is worth noting that there is a lack of consensus on this question in the literature. As we note in the related work, [1] states
>
> >Our experiments show that any random direction $v \in \mathbb{R}^n$  gives rise to similarly interpretable semantic properties. This
>  suggests that the natural basis is not better than a random basis for inspecting the properties of [activation values of some layer]. This puts into question the notion that neural networks disentangle variation factors across coordinates.
>
> It must be noted that in the experiment this quote refers to, the authors look at images in the MNIST validation set maximizing the dot product with a feature vector $v \in \mathbb{R}^n$. This is quite different from the interpretability criterion of [2]. Our intent in highlighting this back-and-forth in prior work is simply to emphasize that when it comes to the question of interpretability of neurons vs. linear combinations thereof, there is significant sensitivity to the details of the experimental setup. And while it is perhaps not shocking the line of reasoning in your review, and our theoretical considerations in sec. 3, our experiment in sec. 6 shows that the presence of nonlinear activations in the hidden layer in question is a crucial experimental detail.
>
> **Skip connections:** yes, that is a better way to put it. We have rephrased that remark!
>
> **Regarding fig. 5 and "5 runs of the experiment":** yes, what we meant was that we trained 10 independently initialized networks and reported distances for 5 of the pairs. In the new version, we do something closer to looking at all $\binom{10}{2}$ pairs: we train 100 models of each architecture to be used in all of the stitching/dissimilarity experiments, and then sample pairs from these ``zoos.'' Details of this strategy can be found in the new sec. D.1. The results obtained appear to be qualitatively similar across the board. We also find the consistency surprising!
>
> **Procrustes:** good point! We now call it a measure.
>
> The "minor points" have been corrected in the new version, thank you for pointing these out!
>
> 1. C. Szegedy et al., “Intriguing properties of neural networks,” arXiv:1312.6199 [cs], Feb. 2014, Accessed: Mar. 24, 2022. [Online]. Available: http://arxiv.org/abs/1312.6199
> 2. D. Bau, B. Zhou, A. Khosla, A. Oliva, and A. Torralba, “Network Dissection: Quantifying Interpretability of Deep Visual Representations,” arXiv:1704.05796 [cs], Apr. 2017, Accessed: Feb. 07, 2022. [Online]. Available: http://arxiv.org/abs/1704.05796

---

### Official Review · Reviewer_kCuW · 2022-07-10

**Rating:** 7
**Confidence:** 4
**Soundness:** 3 good
**Presentation:** 3 good
**Contribution:** 3 good

**Summary:**

The paper contributes to understanding the phenomenon that different networks produce similar performance, which the symmetries can cause.

The paper mainly focuses on ReLU-activated layers, and contributes in the following way: 1) the paper first provides empirical evidence of the similarity of strong model’s representations, then there is a theoretical analysis of the minimal layer for neural stitching that can preserve accuracy—the intertwiner group; 2) the paper proposes a measure of similarity between representations learned with different neural networks, which considers representations have the highest similarity if they differ only up to orthogonal rotation and isotropic scaling; and 3) the paper further analyzes the interpretability of intertwiner groups, though the theory is limited to the certain activation function.


**Questions:**

Please see **Strengths And Weaknesses**.

**Limitations:**

Yes, the authors adequately addressed the limitations and potential negative societal impact of their work.

**Strengths And Weaknesses:**

The paper shows the following **strengths**:

- The topic studied in this paper helps with understanding the neural network better. For now, most of the time, how the neural network works and what representation it extracts remain unclear. This paper contributes to providing a representation similarity measure that would not be affected by orthogonal rotation, which also emphasizes the difference between representations than the baseline does, especially between the lower layer representation and the higher layer representation.

- The paper focuses on ReLU activation, which is a commonly used activation function in deep learning. Thus the result could be helpful in further research in the community.

- The proposed method's efficiency is supported by theoretical analysis and empirical results. Conditions that when theorem (3.3) does not hold are also reported.

- The submission is well organized.

In the meanwhile, I still have the following **questions and concerns**.

- In line 218, the paper mentions the stitching penalty increases up to the third activation layer. The number here (the third layer) is very specific. Would it be too specific to mention this number in conclusion? According to the evidence provided in the paper, I believe there exists such a pattern that the penalties increase first and then decrease, but I am not sure three is a threshold. There are always different network settings, such as the total number of layers and the type of neural network. Different settings might give different thresholds. (To be more clear, I am not saying the experiment is not enough; I am only concerned that it might be too specific to say three is the threshold.)

- The experiment has 5 independent runs, which is acceptable as the confidence interval seems small, though more runs could make the result more convincing.

**Small Things**:

- In line 59, it might be better to add the full name of CKA. I understand that the full name is mentioned in Section 5, but it might be more clear if the full name gets explained as soon as CKA is mentioned.

- In line 197, is it $S(f,\tilde{f}, l, \phi)$?

- In line 259, is it $X-\frac{1}{n}\mathbf{1}\mathbf{1}^T X$ and $Y-\frac{1}{n}\mathbf{1}\mathbf{1}^T Y$?

---

> ### Author Response · Authors · 2022-08-01
> **Clarification of stitching penalty trends and larger sample sizes**
>
> Thank you very much for your thorough read of our paper, as well as your perceptive questions and comments!
>
> **Regarding the trends of stitching penalties** on what was formerly line 218, we completely agree. Even if there appears to be a general pattern that penalties increase then decrease (which seems to hold for the networks we consider, however this is a very small sample), we would not want to claim layer 3 is a threshold in general, only that it seems to be a threshold in our specific set of experiments with the Myrtle CNN, but not for example in our ResNet experiments. We have attempted to clarify that statement in the paper!
>
> **We agreed that a larger sample size would improve the results, and have significantly increased the number of runs.** For example, in our Myrtle CNN stitching experiments we use 32 runs. For the ResNets we were only able to afford 16 runs (the cost of these experiments is linear in the number of layers being stitched and the ResNets have many more than the Myrtle CNN). The dissimilarity measure experiments have also been expanded. In general details on the expanded numbers of runs can be found in the captions of the figures.
>
> To facilitate scaling the experiments, we modified our approach to sampling pairs of models. Rather than training 5 pairs of each architecture, we train 100 models of each architecture and sample pairs with replacement. This is discussed in sec. D.1. We also slightly modified our dataloaders and augmentation to adopt [FFCV](https://ffcv.io/), and as a byproduct the baseline accuracy of our models is slightly higher.
>
> Finally, many thanks for pointing out some not-so-small typos, these have been corrected in the new version!

---

### Official Review · Reviewer_Eay8 · 2022-07-11

**Rating:** 6
**Confidence:** 2
**Soundness:** 3 good
**Presentation:** 2 fair
**Contribution:** 3 good

**Summary:**

This paper introduce an intertwiner group $G_\sigma$ of an activation $\sigma$, which is a subgroup of $GL(n)$ that a set of invertible matrices that commute regarding the activation function, i.e., $A \circ \sigma = \sigma \circ A$ (for, e.g., ReLU). This group structure can naturally explain the weight space symmetry of neural networks, i.e., a realization map $\phi: W \mapsto f$ is $G_\sigma$-invariant s.t. $\phi(W') = \phi(W)$ where $W'$ is a transformed weight by acting a set of elements in $G_\sigma$. The authors use the intertwiner group to investigate the internal representations of ReLU networks from the perspective of the model stitching and representation dissimilarity. The $G_{ReLU}$-based stitching and dissimilarity metric provide a consistent result with previous works. In addition, they provide an application of the intertwiner group for explainable AI.

**Questions:**

For the major comments, please see Strengths and Weaknesses. In short, I believe the presentation of key results and their significance needs improvement.

A minor comment: in line 173 of page 6, it seems that $f_{\leq i}$ should be  $f_{\leq l}$.

**Limitations:**

The authors discuss the limitation of the proposed method in the Limitations section, and I agree with such limitations.

**Strengths And Weaknesses:**

Investigating the learned representation based on symmetry is an interesting problem. The intertwiner group might be a principled group to explain the underlying symmetry behavior of neural networks.

The mathematical concept of the intertwiner group and its property are well-presented and discussed.

The paper is generally well-written, while it is a bit hard to read mainly due to the confusing structure of the paper. For example, Fig. 2 and Fig. 3 appear in the early stage of the paper; however, they are hard to understand before reading the experiments (Section 4 - 6).

Regarding the weight space symmetry, I am not sure whether the introduced intertwiner group is particularly novel or significant compared to previous studies based on other groups, e.g., permutations and scaling. At first glance, the intertwiner group seems to be a comprehensive version of such studies (from the perspective of various activations beyond ReLU, e.g., LeakyReLU, RBF, and polynomial?), but it is not clearly presented in the current version of this paper.

For the model stitching experiment, I agree that the internal representations of separately trained ReLU networks might be invariant up to the intertwiner group. It might be a piece of good evidence or tool when discussing the “Anna Karenina” scenario of deep neural networks. One concern is that the stitching penalty of full 1 x 1 convolution is much lower than that of the intertwiner group, as shown in Fig. 1. It will be nice if the authors clarify it.

From the perspective of the metric, it seems that the intertwiner group provides a tighter bound for hidden layer dissimilarity compared to the previously proposed one (that is based on the orthogonal group). However, I am not sure what is the main use-case of this metric. It would be nice if the authors mention regarding it more precisely.

Overall, to me, the significance of the intertwiner groups compared to the previous work is not entirely clear, although the introduced mathematical concept of them seems to be solid.  Because I am not familiar with the topic of this paper, there might be mistakes in my current evaluation. Please correct me if I am wrong.

---

> ### Author Response · Authors · 2022-08-01
> **Comparing stitching penalties, clarifying novelty of intertwiner groups and motivation for our dissimilarity measures**
>
> Thanks very much for your insightful comments and questions! We will attempt to address each of them in turn:
>
> We agree that Figs. 2-3 are not fully explained at the points where they appear, and have attempted to better describe them in the introduction and captions, as well as provided further pointers to Secs. 4-6 in the captions.
>
> **Comparing stitching penalties:** The stitching penalty of full 1x1 convolution is indeed much lower than that of the intertwiner group. In some sense, this is a good thing: if it were not lower, our results would be in tension with those of [1, 2] as discussed in the second-to-last paragraph of Sec. 4.  On the other hand, we still find it interesting that the intertwiner group accounts for a substantial fraction of the stitching accuracy seen with 1x1 convolutions. We have added a couple sentences on these points to the introduction in an attempt to add clarity/context up front. One last note on this topic: optimization over permutations is non-trivial, and so our results can be viewed as an upper bound on the stitching penalty with intertwiners. It is possible a more sophisticated optimization technique gives lower penalties than we present here (although we made a concerted effort, including a new attempt to chose our negative-L2-regularization multiplier $\alpha$ by cross validation, described in sec. D.5).
>
> **Regarding novelty and significance of the intertwiner group:** we do not claim much concerning the groups listed in Table 1, however we do present a relatively uniform way of calculating these groups for any given activation function, and this did seem to be absent in the literature (in the cases we are aware of network symmetries are simply exhibited, rather than calculated). We have added further commentary under Def. 3.2. Moreover, we would like to emphasize that the novelty/significance of our paper does not hinge on the novelty/significance of the intertwiner group. Rather, theoretical considerations related to intertwiners led motivated us to conduct three new experiments in secs. 4-6. We argue these experiments are significant due to their interesting connections with existing representation learning and interpretability research.
>
> **Regarding use cases of our $G_{\mathrm{ReLU}}$-based dissimilarity measures:** prior work in [3, sec. 2] frames the problem of designing dissimilarity measures in terms of a choice of underlying symmetry group. Our results of secs. 3-4 suggest that the intertwiner group is a natural (and in some sense the smallest possible) choice: the hidden features of two networks with the same architecture could differ by an intertwiner transformation even if the two networks compute the same function from inputs to outputs. This perhaps accounts for the tighter bound on dissimilarity that you point out. Prior work has also compared the results of fully connected stitching experiments with orthogonal CKA (if these are two measures of hidden feature dissimilarity, then assessing whether they give qualitatively similar conclusions serves as a ``sanity test'') [4]. Since our theoretical results motivate experiments stitching with intertwiner groups, we felt it made sense to also compute dissimilarity measures with the intertwiner group as the underlying symmetries.
>
> Finally, you are absolutely right about $f_{\leq l}$.
>
> 1. Y. Li, J. Yosinski, J. Clune, H. Lipson, and J. Hopcroft, “Convergent Learning: Do different neural networks learn the same representations?,” 2015.
> 2. L. Wang et al., “Towards Understanding Learning Representations: To What Extent Do Different Neural Networks Learn the Same Representation,” 2018.
> 3. S. Kornblith, M. Norouzi, H. Lee, and G. E. Hinton, “Similarity of Neural Network Representations Revisited,” ICML, 2019.
> 4. Y. Bansal, P. Nakkiran, and B. Barak, “Revisiting Model Stitching to Compare Neural Representations,” ArXiv, 2021.

---

> > ### Comment · Reviewer_Eay8 · 2022-08-08
> > **Reply to the authors**
> >
> > I appreciate the authors’ thorough response to my questions. Many of the concerns and misunderstood points raised in my review are addressed. Now I think the paper’s contribution outweighs my initial concerns. I raised the review score accordingly.

---

### Author Response · Authors · 2022-08-02
**We thank all the reviewers for their valuable feedback, which we feel has improved our revised paper**

Here we highlight a few key updates.
- The revision includes multiple new experiments: stitching using fully-connected 1-by-1 convolutions with L1 weight decay, a strategy to chose the negative-L2-regularization multiplier with cross validation, stitching with a small vision transformer, and finally dissimilarity measures for a CNN with constant channel width.
- All stitching and dissimilarity measure experiments include more runs, as described in the captions of the figures.
- We have clarified the relationship of our $G_{\mathrm{ReLU}}$-CKA measure to existing versions of Centered Kernel Alignment. It is in fact an instance of CKA, where the kernel is taken to be $k(x, y) = \max(x \odot y)$ -- in our previous version we had mistakenly referred to $G_{\mathrm{ReLU}}$-CKA as a "variant of CKA." We added lem. 5.5, which states that  $\max(x \odot y)$ is a positive semi-definite kernel function (this is proved in the appendix). To the best of our knowledge this kernel has not been previously studied, neither in the context of CKA nor kernels more generally.
- The main body and supplementary are now combined in one document.

---

### Meta-Review · Area_Chair_9pHK · 2022-08-31

**Recommendation:** Accept
**Confidence:** Certain

**Metareview:**

**Summary**: This paper studies symmetries of the space of neural network parameters, i.e. invertible transformations of the parameters which leave the forward function invariant. The authors compute this set of symmetries, called the intertwiner group, for networks with different activation functions. These symmetries have been defined and studied before (as notede by the authors). The primary focus of this work is on the impact of these symmetries on learned hidden representations in the network. The author investigate to what extent networks trained from different random initializations effectively learn the same hidden features, and what amount of variation is due to parameter space symmetry. Through splicing experiments (Section 4), they show that representations from one network can spliced into another using the intertwiner group, which results in only a small drop in performance. In Section 5, the authors compute use two novel metrics to compute how close learned hidden representations are to being related by a single intertwiner for the whole dataset. Section 6 shows, using network dissection, that the significance neuron activations (versus linear combinations of activations) to interpretability depends on the choice of activation function.

**Strengths**: Reviewers [Eay8] and [kCuW] commented that using the intertwiner group to explain weight-space symmetries of neural networks is an interesting and promising approach. Moreover, the paper is well written and well-organized (if difficult to understand). The intertwiner group and its properties are well-presented. Reviewers [kCuW] and [cBdu] further note that the paper presents a good combination of empirical and theoretical results. Discussion of conditions when theorem 3.3 do not hold are reported. Reviewer [kCuW] comments that focusing on ReLU is helpful, since this is a commonly used activation function.

Reviewer [cBdu] comments that while authors are not the first to point out the existence of these symmetries, their treatment is rigorous, more general than in related work, and easy to follow. The reviewers indicates that a contribution relative to related work is that the authors connect weight-space symmetries to the questio of variability of learned representations. The authors achieve good results in splicing experiments with G_{relu}, which in principle require combinatorial optimzation over a permutation group, by using a convex relaxation using doubly-stochastic matrices. Moreover, the appendix and its proofs appears quite thorough.

**Weaknesses**: Reviewer [Eay8] found the paper difficult to read. An example is that early figures cannot be understood until reading the section about experiments. The reviewer also found that it was not sufficiently clear how novel or significant the use of the intertwiner group is relative to previous studies based on permutation and scaling groups.  Other reviewers also had comments on the experiments. Reviewer [kCuW] notes that more reruns for the experiments might be helpful. Reviewer [cBdU] comments that  splicing experiments could be more exhaustive. The reviewer also finds tha the interpretability argument in Section 6 is not that strong, and perhaps beside the point of the paper. Finally, results in Section 5 are hard to interpret without baseline numbers.

**Author Reviewer Discussion**: In response to reviewer [kCuW], the authors performed additional runs  (now 32 in total). In response to reviewer [cBdU] they added experiments on pair-wise comparison for stitching experiments, as well as answering question. The reviewer notes that they appreciate the responses. To address concerns by reviewer [Eay8] the authors improved captions in early figures. They also clarified that in their opinion, the main significance of this paper is not the use of intertwiner group per se (though this does provide a uniform perspective), but that considerations related to intertwiners motivate the experiments in Sec 4-6, which lead to interesting connections with existing representation learning and interpretability research. Reviewer [Eay8] raised their score 5->6 after discussion.

**Reviewer AC Discussion**: Reviewer [cBdU] reiterates that they are happy with the paper and the author responses and that they feel this paper can appear. Other reviewers did not engage, which the AC interprets as a signal that they do not object to the paper appearing.

**Overall Recommendation**: The AC is satisfied with the level of engagement from both reviewers and authors during the review process. While there is no strong champion for this paper, there is also consensus that this paper is above the threshold for acceptance. On this ground the AC considers this a relatively clear accept.

**Award:**

No

---

### Decision · Program_Chairs · 2022-09-14

Accept